# LASeR: Learning to Adaptively Select Reward Models with Multi-Armed Bandits

## Abstract

Reward Models (RMs) play a crucial role in aligning large language models (LLMs) with human preferences, enhancing their performance by ranking outputs during inference or iterative training. However, the degree to which an RM generalizes to new tasks is often not known *a priori*. For instance, some RMs may excel at scoring creative writing, while others specialize in evaluating math reasoning. Therefore, using only one fixed RM while training LLMs can be *suboptimal*. Moreover, optimizing LLMs with multiple RMs simultaneously can be prohibitively computationally-intensive and challenging due to conflicting signals from different RMs, potentially degrading performance. To address these challenges, we introduce **LASeR** (**L**earning to **A**daptively **Se**lect **R**ewards), which iteratively trains LLMs using multiple RMs, selecting and utilizing the most well-suited RM for each instance to rank outputs and generate preference data, framed as a multi-armed bandit problem. Our empirical results on commonsense and math reasoning tasks demonstrate that LASeR can boost iterative LLM optimization by optimizing for multiple RMs, improving the absolute average accuracy of Llama-3-8B over three datasets by $2.67\%$ over training with ensemble RM scores while also showing superior training efficiency (e.g., a $2\times$ speedup). Moreover, on WildChat, a benchmark of instruction-following prompts in open-form generation, we find that using Llama-3-8B LASeR leads to a $71.45\%$ AlpacaEval win rate over sequentially optimizing multiple RMs. Extending to long-context generation tasks, we find that on Llama-3-8B, LASeR achieves an average improvement of $2.64$ F1 points on single-document QA tasks and $2.42$ F1 points on multi-document QA over random RM selection when used with best-of-$n$ sampling. Our analysis shows that LASeR is robust to noisy rewards and generalizes to multiple settings. Finally, we demonstrate that LASeR's RM selection changes depending on the underlying task or instance, and we verify the presence of conflicting preferences from multiple RMs, which can be mitigated using LASeR.

## 1 Introduction

When comparing two responses, human preferences often differ depending on factors like the underlying task, who the annotators are (Santurkar et al., 2023; Ahmadian et al., 2024), and how preferences are elicited (Bansal et al., 2024). Therefore, models of preference data are also likely to differ and might include noise as well as any biases contained in the preference data used to train them. This can pose a problem when using such models as "reward models" (RMs) to align large language models (LLMs) to human preferences using reinforcement learning with human feedback (Christiano et al., 2017; Ziegler et al., 2019; Ouyang et al., 2022). Recent work has focused on aligning LLMs through iterative training, using reward models as proxies for human judgment (Gulcehre et al., 2023), leveraging the LLM to act as an implicit RM or judge (Yuan et al., 2024b; Chen et al., 2024b), or using the gold answer to compute a reward (Pang et al., 2024). Under this paradigm, there are three stages to training LLMs: (i) *generating multiple responses* to a query from an LLM; (ii) *scoring the responses with an RM* to create preference data with better and worse responses; and (iii) using *model-generated preference data* to further train the LLM. Note that for most domains, the gold reward is not readily available, making the quality of the RM or the degree to which it reflects human preferences (i.e., the gold reward) crucial to improving LLM performance. Indeed, several prior efforts aim to train new RMs that better reflect human preferences (Lambert et al., 2024).

However, *selecting one reward model* to guide LLM training can be *suboptimal* for three main reasons: (1) A single RM may not be generalized to heterogeneous sets of examples. RMs are typically designed to reflect specific objectives and may be trained on offline preference datasets. Thus, an RM that performs well on one dataset or domain *may not generalize effectively* to others, leading to misaligned outputs across different tasks or domains (Kirk et al., 2023; Chen et al., 2024a; Casper et al., 2023; Gao et al., 2023). For instance, creativity plays a key role in evaluating the quality of a story, whereas correctness is more important in scoring math solutions. (2) RM performance leaderboards (e.g., Lambert et al. (2024)) that rely on human-annotated preferences can have unreliable rankings due to the presence of *incorrect and ambiguous preferences* (Yu et al., 2024; Hejna et al., 2023). (3) Lastly, over-optimization on a particular RM can lead to *reward hacking* (Skalse et al., 2022; Rafailov et al., 2024a), resulting in minimal gain or even drops in downstream performance.

To mitigate these issues, a prevalent approach is to *ensemble multiple reward models* (Coste et al., 2023; Eisenstein et al., 2023; Zhang et al., 2024; Ramé et al., 2024). However, these methods also come with significant challenges: as RMs are typically based on LLMs, training with multiple RMs often requires loading and managing several large models simultaneously, which can be *computationally expensive*, becoming infeasible as models increase in size. Moreover, aggregating multiple RM scores together is susceptible to *noisy rewards or conflicting preferences* from RMs, especially RMs that are not well-suited for the specific task (Rita et al., 2024). This, in turn, can degrade the quality of the preference data, leading to low-quality updates during training (Wang et al., 2024a). Finally, manually selecting a subset of RMs to combine is a labor-intensive process that involves training many different variants on a combinatorially large set of RM groupings. This underscores the need for more efficient methods that efficiently and robustly optimize LLMs using multiple RMs.

In this work, we introduce **L**earning to **A**daptively **Se**lect **R**ewards (LASER), that, given a set of RMs, adaptively and efficiently *selects* a suitable RM for each instance by casting selection as a *multi-armed bandit* problem (Vermorel & Mohri, 2005; Audibert et al., 2009). Specifically, during training, the RM (arm) is chosen dynamically based on contextual information about the model's performance and past interactions. The LLM is then fine-tuned based on the RM-annotated data, and the bandit's parameters are updated accordingly to reflect the performance of the LLM after training on preference data annotated using selected RM (see Fig. 1). By design, LASER's adaptive instance-level RM selection (c.f. Sec. 3) addresses the three shortcomings of choosing one reward model (lack of generalization, unreliable rankings, and over-optimization) and outperforms using the same RM across all instances, yielding higher downstream performance and better generalization. Moreover, unlike previous multi-RM methods that require simultaneously loading and running multiple RMs (Ramé et al., 2024; Coste et al., 2023), our method selects one reward model at a time (Sec. 4). This makes the training more efficient and improves performance by allowing the model to adaptively focus on the most suitable RM for each specific instance or phase of training.

Empirically, we demonstrate the effectiveness of LASER for iteratively training LLMs using multiple RMs on three broad domains: reasoning, instruction-following in text generation, and long-context understanding (Sec. 4.2). We show that on reasoning benchmarks such as StrategyQA (Geva et al., 2021) (testing commonsense reasoning) and GSM8K (Cobbe et al., 2021) (testing math reasoning), LASER with Llama-3-8B improves absolute accuracy (averaged across 3 datasets) by 1.45% over the best single RM and 2.67% over an ensemble of RM scores. With Mistral-7B, LASER outperforms RM agreement ensemble baseline by 1.65% in absolute accuracy. LASER is also effective on general instruction-following: we show that using LASER with four strong 7B RMs from RewardBench to finetune Llama-3-8B on a subset of WildChat (Zhao et al., 2024) beats LLMs trained with the best RM in the RM score ensemble and a sequential baseline, with 56.34% and 71.45% win rates (respectively) on length-controlled AlpacaEval (Dubois et al., 2024). LASER also beats the RM score ensemble, with 72.69% and 73.27% win rates using Llama-3-8B and Mistral-7B. Moreover, our results show the effectiveness of LASER's RM selection strategy at inference-time on long-form generation tasks; on LongBench (Bai et al., 2022), we find LASER beats random RM selection baseline by 2.64 F1 points on single-document QA tasks and 2.42 F1 points on multi-document QA when using best-of-$n$ sampling for Llama-3-8B. Our analysis reveals that LASER is more efficient than sequential multi-RM and RM ensemble baselines in terms of training time (wall-clock hours) by a factor of $3\times$, and $2\times$, respectively while being more robust to noisy rewards and conflicting preferences from multiple RMs (Sec. 5). Finally, we demonstrate that LASER effectively selects RMs based on the underlying instance and generalizes to multiple settings, including out-of-distribution datasets, different training loss functions, etc.

## 2 RELATED WORK

**Multiple Reward Ensembles.** Training large language models (LLMs) with multiple reward functions is an emerging research area focused on aligning model outputs with complex objectives that require diverse evaluation metrics. One prior approach involves using ensembles of multiple rewards (Ramé et al., 2024; Wu et al., 2024; Coste et al., 2023; Zhang et al., 2024; Wang et al., 2024b; Jang et al., 2023; Eisenstein et al., 2023). Unlike these methods, which train multiple RMs or use multiple RMs during LLM training, LASER selects only a single pretrained RM at each LLM training step. Not only is optimizing for one RM at a time more efficient, but it also avoids the problem of having conflicting rewards (Rita et al., 2024). In Sec. 4.2, we show that LASER outperforms multiple variants of optimizing LLMs with an ensemble of RMs aggregating scores (consistent with Coste et al. (2023)) and based on the agreement between preferences from different RMs (similar to Wang et al. (2024d)). Another line of work uses mixture-of-experts (MoE) techniques for training an RM from multiple interpretable objectives (Wang et al., 2024c) or jointly training task-specific RMs and a sparse MoE router (Quan, 2024). Instead of relying on static datasets with human-annotated attributes for RM training (as Wang et al. (2024c) does), we employ existing RMs to train LLMs on its own generations, a strategy that has been shown to be more effective (Ivison et al., 2024). Unlike Quan (2024), who jointly train multiple RMs and sparse MoE router (requiring loading all models at once), LASER implicitly and efficiently learns a sparse router and does not need to train RMs. LASER performs RM selection via multi-armed bandits over existing off-the-shelf RMs with strong performance on leaderboards (Lambert et al., 2024).

**Iterative LLM Training.** Recent works on training LLMs incorporate reinforcement learning with human feedback (RLHF) due to its effectiveness in improving instruction following the ability of LLMs over their pretrained counterparts (Ouyang et al., 2022; Bai et al., 2022; Touvron et al., 2023). The standard RLHF training pipeline, which comprises finetuning the LLM on static datasets with human-annotated preferences, is bottlenecked by the size and quality of annotated preference data as well as the limited effectiveness of off-policy optimization (Xu et al., 2023; Xiong et al., 2024; Yuan et al., 2024b; Guo et al., 2024). To remedy this, recent work focuses on training LLMs iteratively, scoring the LLM's generations to create feedback data for RLHF. This line of work obtains scores either from gold labels (Singh et al., 2023; Pang et al., 2024), from a single RM (Gulcehre et al., 2023), or from the generating LLM (Yuan et al., 2024b; Chen et al., 2024b). In contrast, we take advantage of the abundance of publicly-available RMs and growing interest in developing RMs (Lambert et al., 2024) by using *multiple* RMs. This has several advantages over past work: it works in cases where gold labels are not available (e.g., generation tasks), deals with the three issues associated with using a single RM (unreliable rankings, lack of generalizability, and over-optimization), reduces the burden on the user to pick the right RM, and avoids problems stemming from the inability of LLMs to judge their own responses for certain domains (Huang et al., 2023).

## 3 LASER: **L**EARNING TO **A**DAPTIVELY **SE**LECT **R**EWARDS

In this section, we describe LASER in detail. First, we expand on the training pipeline (in one iteration) with a general reward function (Sec. 3.1). Then, in Sec. 3.2, we describe how LASER dynamically selects an RM from a set of multiple RMs using MAB algorithms, i.e., how we dynamically assign the reward function for a given instance or batch of instances. Finally, in Sec. 3.3, we describe the overall training setup across iterations and specifically how we update the parameters of the MAB at the end of each iteration. A detailed illustration of LASER is shown in Fig. 1.

### 3.1 TRAINING LLMS USING A REWARD FUNCTION

LASER involves training with multiple RMs using a multi-arm bandit (MAB), which selects one model at a time. Therefore, we first describe how we train LLMs with generated data assuming a single RM; this corresponds to the top-right in Fig. 1 (in blue).

**Notation.** Following Yuan et al. (2024b) and Pang et al. (2024), we adopt an iterative training pipeline to finetune the LLM for $M$ iterations. Let $\pi_m$ be the LLM at iteration $m$; we assume that we start from an initial pretrained model $\pi_0$. Let $\mathcal{D} = \{x_1, x_2, \ldots, x_N\}$ represent the training inputs, where $x_i$ is an input query or prompt. Corresponding to each input query $x_i$, we sample a

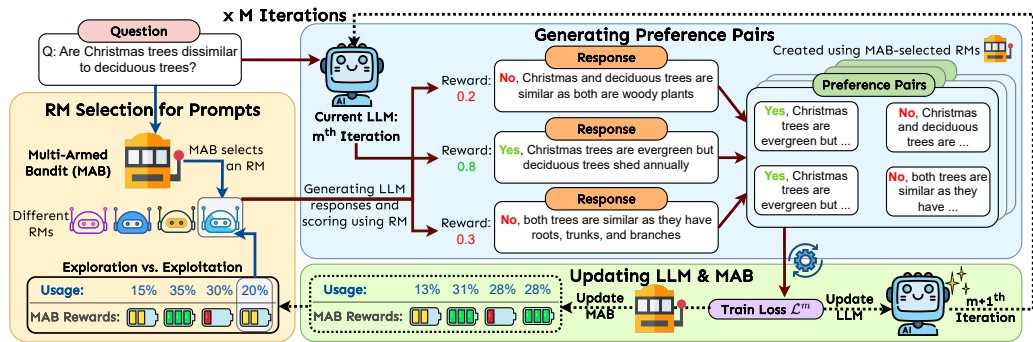

Figure 1: Overview of LASER. Given the query, the multi-armed bandit selects an RM depending on the underlying query and the bandit's parameters (based on the usage of each RM and the expected MAB reward). At iteration $m$, the LLM generates multiple responses that are scored based on the selected RM for that query scoring each response. These responses are ranked into preference pairs, which are then used to fine-tune the model. The same train loss $\mathcal{L}^m$ is used to update the parameters of the LLM as well as the MAB for the next iteration, making the entire pipeline iterative.

set of $n$ responses from the LLM at the current $m^{\text{th}}$ iteration as $\mathbf{y}_i = \{y_i^1, y_i^2, \ldots, y_i^n\} \sim \pi_m(y|x_i)$. Let $R^\star : (y_i^j | x_i) \to \mathbb{R}$ be a reward function that can score an LLM generated response $y_i^j$ to a query $x_i$ based on how well it aligns with specific task objectives or instructions. Note that $R^\star(.)$ can be any reward function and may correspond to a single RM, one of the multiple RMs selected by the MAB (as in our case), or even the true reward.

**Generating Preference Pairs.** We evaluate each response $y_i^j$ using the reward function $R^\star(y_i^j | x_i)$. By comparing the rewards assigned to different responses, we can form $P$ preference pairs $(y_i^w, y_i^l)$, where $y_i^w$ is preferred over $y_i^l$ if $R^\star(y_i^w | x_i) > R^\star(y_i^l | x_i)$, thereby building a preference dataset:[1]

$$\mathcal{D}_{\text{pref}} = \{(x_i, y_i^w, y_i^l) \mid x_i \in \mathcal{D}, R^\star(y_i^w) > R^\star(y_i^l)\}.$$

**Training Loss Function ($\mathcal{L}^m$).** In each iteration, we fine-tune the model using the generated preference dataset $\mathcal{D}_{\text{pref}}$, resulting in $M$ models $\pi_1, \pi_2, \ldots, \pi_M$. Specifically, we update the model using the DPO loss (Rafailov et al., 2024b) for learning from the preference pairs. In this work, we use the following loss functions for training the LLM at iteration $m$:

$$\mathcal{L}_{\text{DPO}}^m(\pi_m) = -\mathbb{E}_{(x_i, y_i^w, y_i^l) \sim \mathcal{D}_{\text{pref}}} \left[ \log \sigma \left( \beta \log \frac{\pi_m(y_i^w \mid x_i)}{\pi_{m-1}(y_i^w \mid x_i)} - \beta \log \frac{\pi_m(y_i^l \mid x_i)}{\pi_{m-1}(y_i^l \mid x_i)} \right) \right]$$

$$\mathcal{L}_{\text{NLL}}^m(\pi_m) = -\mathbb{E}_{(x_i, y_i^w) \sim \mathcal{D}_{\text{pref}}} \left[ \frac{\log \pi_m(y_i^w \mid x_i)}{|y_i^w|} \right], \tag{1}$$

where $\pi_m$, and $\pi_{m-1}$ denotes the LLM in the current iteration $m$ and the previous iteration $m-1$ (used as the reference model in DPO loss). Following Yuan et al. (2024b), we use the standard DPO loss for instruction-finetuning. Following Pang et al. (2024), we use the NLL loss on the preferred responses as an additional regularizer for reasoning tasks, i.e., $\mathcal{L}^m = \mathcal{L}_{\text{DPO}}^m + \mathcal{L}_{\text{NLL}}^m$. In Appendix C, we show that LASER outperforms baselines irrespective of the choice of the loss function $\mathcal{L}^m$.

### 3.2 BANDIT ALGORITHMS FOR ADAPTIVE RM SELECTION

Sec. 3.1 described the data creation and LLM training procedure for our method when using a general RM (Fig. 1; top-right), which trains the LLM for a single mini-batch. Here, we describe the process by which we adaptively *select an RM for each batch of queries* using bandit algorithms (shown in Fig. 1-left, in yellow) and update the parameters of the bandit (more details in Appendix A.2).

---

[1]Following Pang et al. (2024), we randomly sample $P = 10$ pairs corresponding to each prompt $x_i$. For brevity, we omit this in the notation of $\mathcal{D}_{\text{pref}}$; but in our setting $|\mathcal{D}_{\text{pref}}| = P \times |\mathcal{D}|$.

**Background: Multi-Armed Bandits.** The multi-armed bandit (MAB) problem addresses the challenge of balancing exploration and exploitation in sequential decision-making (Vermorel & Mohri, 2005; Audibert et al., 2009). The goal is to maximize cumulative MAB rewards over time by selecting arms that yield the highest MAB rewards.[2] A decision-making agent faces a trade-off: whether to exploit the arm with the highest known MAB reward based on past observations or explore other, less familiar arms to gather more information that might lead to even better rewards in the future. In a contextual MAB setting, the agent is also provided with additional information in the form of a context, such as current state and input, to help inform arm selection accordingly. LASER uses MABs to dynamically identify the most suitable RM for each query $x_i$ through exploration while simultaneously training the LLM. Pulling a previously un(der)-explored arm allows the MAB to update its information about the relevance and quality of preference pairs built using that RM via the MAB reward (discussed below).

**RM selection in LASER.** LASER uses mini-batch training for each iteration, i.e., we use MABs to select a *single RM for a batch of prompts* $\mathbf{x}_{m,t}$ for $t^{\text{th}}$ batch or training step of iteration $m$ (total of $T$ steps/batches in each iteration).[3] Let the set of $K$ reward models (or arms) denoted by $\mathcal{R} = \{R_1, R_2, \ldots, R_K\}$, where each $R_k$ corresponds to a different RM. We employ `LinUCB` (Li et al., 2010), a contextual bandit algorithm for the arm or RM selection. We choose LinUCB because it is a contextual bandit algorithm (i.e., it takes into account the context information), is easy to incorporate into our framework, and provides a good trade-off between computational efficiency and performance. Li et al. (2010) assume that the MAB reward – in our case, the cumulative train loss function on the batch ($\mathcal{L}^m$) at the given iteration $m$ – can be modeled linearly as a function of context features and computes the expected MAB reward of each arm with an upper confidence bound to ensure exploration (Garivier & Moulines, 2008; 2011). In each step $t$, we have a batch of input prompts $\mathbf{x}_{m,t}$ for which we compute sentence embeddings, using the policy model $\pi_m$, and use the mean sentence embedding as the context $c(t)$ to the MAB, i.e., $c(t) = \sum_{x \in \mathbf{x}_{m,t}} e_m(x)/|\mathbf{x}_{m,t}|$, where $e_m(.) \in \mathbb{R}^d$ yields the sentence embedding from the model $\pi_m$. We calculate the embedding for a prompt as the last token embedding from model $\pi_m$ (details in Appendix A.1). The learned parameters of the LinUCB bandit include $\hat{\theta}_k \in \mathbb{R}^d$ which represents the learned weights for the features of each reward model and $A_k \in \mathbb{R}^{d \times d}$ (a covariance matrix) and a bias vector $b_k \in \mathbb{R}^d$ corresponding to each arm or RM $R_k$. We initialize the parameters for LinUCB by randomly initializing $b_k$ and setting parameter $A_k$ to the identity matrix. Based on the LinUCB algorithm, for each batch, the selected RM $R_t^\star$ is determined by:

$$R_t^\star = R_j, \text{ such that } j = \arg\max_{k \in [1,K]} \left( c(t)^\top \hat{\theta}_k + \alpha \sqrt{c(t)^\top A_k^{-1} c(t)} \right), \tag{2}$$

where $\hat{\theta}_k = A_k^{-1} b_k$. $A_k$ and $b_k$ are updated based on the MAB reward for each RM, which corresponds to the normalized negative train loss $-\hat{\mathcal{L}}^m$ (described in detail in Appendix A.2):

$$A_k \leftarrow A_k + c(t)c(t)^\top; b_k \leftarrow b_k - \hat{\mathcal{L}}^m(t)c(t). \tag{3}$$

## 3.3 LLM AND BANDIT TRAINING IN LASER

A key aspect of our approach is the generation of new preference training data in each iteration using the generations of the LLM itself and the RM selected by the MAB. Fig. 1 presents our training procedure, broken down into three stages: (i) the MAB selects an RM $R_t^\star$ (see Sec. 3.2; Fig. 1 left), generating preference pairs by scoring the LLM's outputs using the RM (Fig. 1 (top-right)), and parameter updates to the LLM and MAB. In this way, the model continuously learns from its own outputs, guided by the selected reward model. After each LLM train step (i.e., one mini-batch), the MAB's parameters are updated based on the observed MAB reward, i.e., how much the LLM's loss decreased from using the selected RM. In the case of LinUCB, this involves updating the parameter estimates $b_k, A_k$ (see Fig. 1; bottom in green). This entire process – selection of reward

---

[2]In order to distinguish between rewards or scores generated by RMs and the rewards used in MAB literature, we refer to the latter as "MAB rewards".

[3]Note that LASER can switch between RMs at the instance level if the batch size is set to 1; however, for the sake of efficiency, we batch instances together both for LASER and the baselines, as this reduces computational overhead associated with loading RMs onto the GPU.

models, generation of new supervision data, fine-tuning, and bandit updates – repeats for a total of $M$ iterations (summarized in Algorithm 1).

**LASER with Best-of-$n$ Sampling.** For settings where finetuning the LLM is not desirable or feasible, LASER can also be applied to learn the MAB parameters without training the LLM. Rather than fine-tuning the model with preference data, we employ best-of-$n$ sampling (Lightman et al., 2023; Sun et al., 2024), where multiple responses are generated, and the best one is selected based on the RM. The bandit parameters are then updated using equation (3), with the MAB reward calculated as the negative normalized NLL loss on the train data. This updated bandit can subsequently be used for inference on the test set. This approach is particularly useful for tasks such as long-context understanding, where training would be too computationally intensive (example setting in Sec. 4.2).

## 4 EXPERIMENTS AND RESULTS

### 4.1 EXPERIMENTAL SETUP

**Models.** We conduct our experiments on the Llama-3-8B base (AI@Meta, 2024) and the Mistral-7b-v3-instruct (Jiang et al., 2023) models. For training, all models are fine-tuned using Low-Rank Adaptation (LoRA) (Hu et al., 2021) for efficiency. For both training and inference, we do 0-shot prompting and sampling $n = 30$ responses per prompt with temperature 0.8 (see Appendix A.1).

**Reward models.** We select $K = 4$ strong 7B RMs from RewardBench (Lambert et al., 2024), which include Zephyr-7B-Alpha, Qwen1.5-7B-Chat, Eurus-7B-KTO, and OLMo-7B-Instruct. Following the pipeline outlined in Lambert et al. (2024), for these models, we compute the reward for each response as the log likelihood of the RM for that response (details in Appendix A.1).

**Datasets and Metrics.** Our experiments cover a range of tasks and datasets (see Appendix A.1):

- **Reasoning:** Evaluating reasoning abilities is crucial for testing the model's capacity to handle complex, multi-step tasks and has presented a challenge to iterative preference optimization methods (Yuan et al., 2024b; Chen et al., 2024b). We train and evaluate on StrategyQA (Geva et al., 2021), MMLU (Hendrycks et al., 2021b;a), and GSM8K (Cobbe et al., 2021).
- **Instruction-Following:** We further evaluate our method on heterogeneous tasks without gold labels. We use user prompts from WildChat dataset (Zhao et al., 2024), which contains a collection of natural user-chatbot interactions. This dataset has five primary categories of instruction-following prompts: creative writing, analysis, coding, factual information, and math reasoning. Due to computational constraints, we randomly subsample 5K prompts from each category for model training. We compare models trained with LASER against baselines (described below) using length-controlled AlpacaEval (Dubois et al., 2024) that pairs responses from two different LLMs and uses GPT-4 as a judge to pick the winner, accounting for the length of both responses.
- **Long-Context Understanding:** As finetuning LLMs on long-context inputs is computationally intensive, we demonstrate the effectiveness of LASER using Best-of-$n$ sampling on Long-Bench (Bai et al., 2023) which consists of multiple tasks, such as single-document QA, multi-document QA, summarization, and few-shot learning. For the QA and few-shot learning tasks, we measure performance with F1 score, while for summarization we use Rouge-L (Lin, 2004).

**Baselines.** We compare our models against the following baselines:

- **Best RM**: From our collection of RMs, we pick the RM that corresponds to the best overall score on RewardBench (Lambert et al., 2024): Zephyr-7B-Alpha. We use this *single* RM during training (c.f. Sec. 3.1). This baseline reflects the performance gain a user could expect when selecting the best RM from a leaderboard without knowing *a priori* how it generalizes to a particular domain.
- **Avg. RM**: Here, we perform single RM training over all the RMs in our collection and report the average performance. A comparison with this baseline represents an expected gain from a randomly-picked RM from a leaderboard.
- **Random RM Selection:** In this baseline, we randomly sample a single RM from the set of RMs (from a uniform distribution) for each training batch in every iteration. This comparison demonstrates whether, without prior knowledge of which RM is best suited for a downstream task, LASER can outperform random sampling of RMs during training.

Table 1: Performance on reasoning benchmarks. The baselines also include supervised fine-tuning on human-written responses (SFT) as a reference for performance without preference optimization. The highest accuracy is shown in bold, and the second-highest accuracy is underlined. Across both Llama-3-8B and Mistral-7B models, LASER yields the highest accuracy for each task.

| Method | Llama-3-8B | | | | Mistral-7B | | | |
|---|---|---|---|---|---|---|---|---|
| | StrategyQA | GSM8K | MMLU | Avg. | StrategyQA | GSM8K | MMLU | Avg. |
| SFT | 80.41 | 69.43 | 65.66 | 71.83 | 68.57 | 43.62 | 56.48 | 56.22 |
| Best RM | 84.29 | 73.16 | 67.15 | 74.87 | 70.06 | 45.81 | 62.04 | 59.30 |
| Avg. RM | 82.62 | 71.57 | 66.67 | 73.62 | 69.62 | 45.47 | 59.58 | 58.22 |
| Random RM Selection | 84.37 | 71.99 | 67.85 | 74.74 | 69.97 | 45.12 | 59.88 | 58.32 |
| Sequential RM Selection | 83.90 | 72.94 | 68.02 | 74.95 | 70.59 | 46.11 | 59.66 | 58.79 |
| Classifier Selection | 83.13 | 72.73 | 67.96 | 74.60 | 70.31 | 45.28 | 60.35 | 58.65 |
| RM Score Ensemble | 82.96 | 70.94 | 67.04 | 73.65 | 68.89 | 44.53 | 58.23 | 57.22 |
| RM Agreement Ensemble | 84.03 | 73.85 | 68.35 | 75.41 | 70.26 | 45.92 | 61.09 | 59.09 |
| LASER (Ours) | 85.96 | 74.75 | 68.24 | 76.32 | 73.06 | 46.89 | 62.27 | 60.74 |

- **Sequential RM Selection**: In training, this method explores different RM sequentially and based on a set order in each iteration to examine their impact on model training, demonstrating that, instead of optimizing with all RMs, LASER can adaptively select the best RM for each batch.
- **RM Score Ensemble:** We generate multiple responses for each query, which are scored (offline) using each RM, and the preference dataset is created by averaging the scores across all RMs (following Coste et al. (2023)); thus, comparing LASER with using all RMs simultaneously.
- **RM Agreement Ensemble:** Because ensembling scores through averaging is sensitive to the absolute scores produced (which may differ between RMs), we follow Wang et al. (2024d) in ensembling through ranking and agreement. Specifically, we generate 32 responses for each query and sample 100 pairs from each set of 32. We score each pair with each RM, constructing a preference dataset by choosing the 10 pairs for each query with the highest agreement of preference rankings across RMs.
- **Classifier Selection:** To measure compare against a context-sensitive baseline that does not use a MAB, we train a $K$-way classifier to perform RM selection using data from RewardBench (see Appendix A.1). Specifically, for each query and RM, we compute the RM's score of the annotated preferred and disprefered response. The RM that assigns the correct preference ordering with the highest difference between the scores of the preferred and dispreferred responses is chosen as the *RM label* and used to train the classifier. While training the LLM, for each training input $x_i \in \mathcal{D}$, we select the RM for building preference pairs based on this trained classifier.

Conceptually, the best RM baseline serves as an "exploit-only" setting that exploits the best available RM based on aggregate RewardBench scores. On the other hand, the random and sequential selection baselines are "explore-only" in that they pick a new RM either randomly or via a predefined sequence irrespective of the performance of each arm (RM). We train models for each baseline to convergence. In particular, LASER, "Best RM", "Avg. RM", and RM ensemble baselines were trained for 10 iterations. For both the sequential and random RM selection, we found LLM training took longer to converge, and consequently, the model was trained for 25 iterations. The number of iterations for each approach was chosen based on performance on the dev set (see Appendix A.1).

## 4.2 MAIN RESULTS

**LASER achieves the best average accuracy on reasoning tasks.** Table 1 shows our method consistently outperforms the baselines across multiple benchmarks, particularly in the StrategyQA and GSM8K tasks. For example, using the Llama-3-8B model, LASER yields the highest reasoning accuracy average across all tasks, with improvements of approximately 2% absolute accuracy over the sequential baseline on both GSM8K and StrategyQA. On the Mistral-7B model, LASER also improves average accuracy by roughly 2% over the sequential baseline. Additionally, our method outperforms the best RM (based on RewardBench) baseline in average accuracy by 1.45% and 1.44% absolute accuracy with Llama-3-8B and Mistral-7B, respectively. In cases where the best RM is not known beforehand, LASER surpasses the performance of the average RM by 2.7% on Llama-3-8B and using the RM Score Ensemble for each instance by 2.67% and 3.52% on Llama-

Figure 2: Length-controlled AlpacaEval win rates comparing LASeR against baselines on the instruction-following tasks on WildChat using Llama-3-8B and Mistral-7B.

Table 2: LASeR outperforms baselines in long-context understanding tasks with Llama-3-8B and Mistral-7B. Sequential RM selection is not applicable in this setting as only inference is conducted. For QA and few-shot learning tasks, we report F1 scores, and for summarization, we report Rouge-L.

| Method | Single-Doc QA | | Multi-Doc QA | | Summarization | | Few-shot Learning | |
|---|---|---|---|---|---|---|---|---|
| | Llama-3 | Mistral | Llama-3 | Mistral | Llama-3 | Mistral | Llama-3 | Mistral |
| Base model | 33.89 | 26.01 | 32.96 | 24.06 | 29.54 | 26.47 | 70.23 | 64.93 |
| Best RM | 35.12 | 28.93 | 35.83 | **27.93** | **34.26** | **30.42** | 71.79 | 68.34 |
| Random RM Selection | 34.83 | 27.44 | 35.19 | 25.38 | 31.57 | 27.19 | 70.91 | 66.72 |
| RM Score Ensemble | 34.51 | 26.75 | 35.52 | 25.71 | 32.38 | 28.17 | 70.34 | 66.97 |
| LASeR (Ours) | **37.47** | **29.14** | **36.94** | 27.80 | 34.13 | 30.08 | **73.31** | **68.47** |

3-8B and Mistral-7B, respectively (in accuracy averaged over the three tasks). Moreover, this lower performance by the RM Score baseline is not purely due to variance in the scores: LASeR also surpasses the RM Agreement Ensemble by $0.91\%$ and $1.65\%$ on Llama-3-8B and Mistral-7B. Additionally, compared to using a frozen classifier for RM selection, training with LASeR improves average reasoning performance by $1.72\%$ for Llama-3-8B and by $2.09\%$ for Mistral-7B. Overall, LASeR provides *consistent* results while the underlined second-place models show *inconsistent* performance across datasets and models. These results also emphasize the benefit of LASeR, as it eliminates the need to choose a different RM in advance or ensemble multiple RMs.

**LASeR beats baselines at instruction-following.** Often, LLMs are used by large numbers of people with a diverse set of queries, goals, and intentions, and their preferences vary based on the underlying query. To demonstrate the effectiveness of LASeR in such settings, we compare the instruction-following performance in Fig. 2, i.e., AlpacaEval win rates, of LLMs trained using LASeR with the baselines using WildChat. Specifically, with Llama-3-8B, LASeR achieves substantial win rates compared to the random and sequential baselines, with $78.33\%$, and $71.45\%$, respectively. We also outperform training with the single best RM (per RewardBench) by a $56.34\%$ win rate. We hypothesize the lower win rate of the baselines stems from the inability of these baselines to deal with conflicting signals from multiple RMs (see Fig. 5 for further analysis). The results are also applicable to the Mistral-7B model, which achieved a win rate of $58.72\%$ against the best-RM baseline and a win rate of $63.72\%$ against the sequential selection baseline. Lastly, across models, LASeR outperforms classifier-based RM selection (by a win rate of at least $60.37\%$) and both Score and Agreement-based variants of RM ensembling by a win rate of at least $72.69\%$ and $52.64\%$, respectively. Overall, these results highlight that LASeR excels in tasks without gold labels and performs consistently well at following instructions across various user queries, showcasing its adaptability to diverse tasks.

**LASeR's adaptive RM selection helps long-context understanding.** Given the cost of training long-context systems, for LongBench (Bai et al., 2023), rather than finetuning a model using RMs, we employ the selected RM to rerank generation in Best-of-$n$ sampling (see Sec. 3.3). In Table 2, we observe that LASeR consistently outperforms the baselines across tasks on Llama-3-8B and Mistral-7B except on summarization, where we achieve comparable performance. LASeR improves single-doc QA by $3.58$ F1 points over the base Llama-3-8B model and $2.64$ F1 points over random RM selection. On multi-doc QA, our approach improves performance over the Llama-3-8B and Mistral-7B models by $\approx 4$ F1 points each, beating out random RM selection by $2.42$ F1 points on Mistral-7B. Furthermore, on few-shot learning tasks, LASeR provides over 3 points gain in F1

compared to the base model for both Llama-3-8B and Mistral-7B, surpassing the average RM performance by up to 2.4 F1 points (on Llama-3-8B) and demonstrating its effectiveness across tasks. Lastly, Table 2 demonstrates that LASᴇR consistently outperforms the RM Score Ensemble baseline across different long-context tasks and LLMs, e.g. a $\approx 3$ F1 point improvement on single-doc QA and few-shot learning tasks with the Llama-3-8B model.

# 5 ADDITIONAL ANALYSIS OF LASᴇR

**Robustness to Noisy Rewards.** To examine the robustness of our method in the presence of noisy or irrelevant rewards, we conduct the following analysis using Llama-3-8B on GSM8K. We add varying amounts of Gaussian noise $\sigma$ to the rewards generated by RMs sampled from the distribution $\mathcal{N}(0, \sigma I)$, where $I$ is the identity matrix, to simulate noisy rewards when using RMs in out-of-distribution settings. In addition to LASᴇR using the LinUCB algorithm (c.f. Sec. 3.2), we also use Exp3 (Auer et al., 2002) designed for adversarial bandit settings. In Fig. 3, we find that even

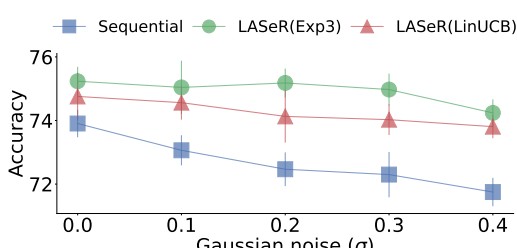

Figure 3: Impact of the magnitude of Gaussian noise on the accuracy of LASᴇR and sequential baseline on RewardBench.

as the degree of noise in RM scores increases (from $\sigma = 0.1$ to $0.4$), LASᴇR's selection strategy continues to perform robustly, mitigating the effects of noise compared to the sequential baseline. Specifically, LASᴇR has an average performance drop of only $0.55\%$ accuracy at a noise level of $\sigma = 0.3$, whereas the sequential baseline suffers a $1.6\%$ accuracy drop (3 times as much) under the same conditions. Furthermore, LASᴇR using Exp3, the most noise-robust method, maintains consistent performance, with only a $0.26\%$ accuracy drop.

**Training Efficiency of LASᴇR.** As we noted in Sec. 4.1, standard multi-reward baselines such as sequential and random RM selection are slow to converge. We now concretely show the accuracy-training time tradeoff in Fig. 4 by comparing the GSM8K performance of training with LASᴇR and different baselines, along with the corresponding wall clock training time.[4] We find that sequentially optimizing over each RM performs the worst in terms of training time ($3\times$ of LASᴇR) while RM score ensemble has the worst accuracy (and takes $2\times$ the training time of LASᴇR). Moreover, LASᴇR outperforms all baselines in terms of accuracy while maintaining the lowest training time, being more than twice as fast as the second-best baseline, RM Agreement Ensemble.

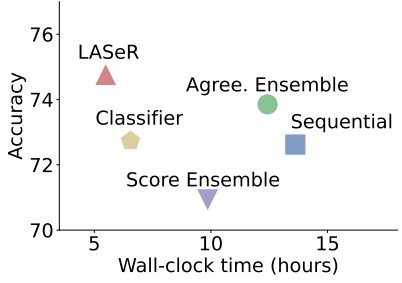

Figure 4: Training efficiency of LASᴇR vs. different baselines on GSM8K.

**Presence of Conflicting Signals among RMs.** In Sec. 4.2, we find that LASᴇR consistently outperforms other multi-reward baselines across a wide variety of tasks. We attribute some of these performance gains to the inability of the multi-reward baseline to handle conflicting signals, resulting in subpar training data from multiple RMs. To study this, we sample pairs of outputs generated by Llama-3-8B on MMLU as well as WildChat and evaluate the consistency of response preferences measured by multiple RMs. Since pair-wise preferences are binary, we compute F1 to measure consistency with one RM's preferences serving as the reference. Fig. 5 (left on MMLU) reveals that Qwen and Zephyr

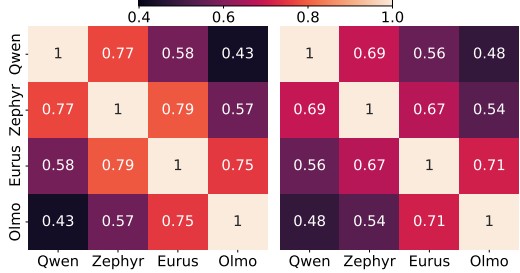

Figure 5: Agreement in preference rankings of Llama-3-8B responses between RMs on MMLU (left) and WildChat (right).

---

[4]Wall clock time is measured as the training time of a model (hours), keeping compute resource consistent.

have the highest agreement rate at $0.77$, while Qwen's agreement with Eurus and Olmo is lower at $0.58$ and $0.43$, respectively. Zephyr also shows low agreement with Olmo at $0.57$. This is expected as Qwen and Zephyr are the top-performing models in reasoning according to RewardBench, while Olmo ranks the lowest in reasoning ability among the four models. We observe similar trends in agreement across RMs on WildChat (albeit with different agreement scores), which contains user queries asked LLMs in the wild; see Fig. 5 (right). It appears that for more heterogeneous datasets with more categories, the level of disagreement among RMs (or conflict) increases. This also highlights LASER's advantages over multi-RM baselines that do not address conflicts in RMs and may explain why choosing one RM in LASER and best RM baseline outperforms multi-RM ensembles.

**LASER's selected RM adjusts to the Query.** Fig. 6 shows the relative utilization rates of each arm (i.e., RM) of the bandit on WildChat.

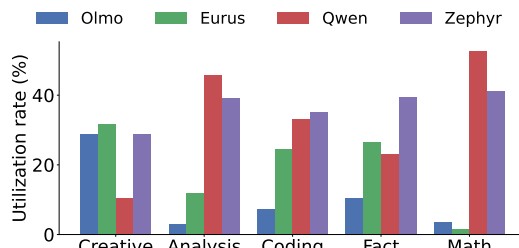

We observed vastly different RM utilization rates depending on the underlying query within the *same dataset*. We observe a similar trend in LASER's RM selection on LongBench in Appendix B (refer to Fig. 7). On queries requiring creativity in LLM responses, we find that Olmo and Eurus RMs are utilized about 20% more often than Qwen RM, despite Qwen RM being ranked higher on RewardBench. This can be explained by the fact that the Qwen RM largely underperforms on the "chat" subsplit of RewardBench (behind Olmo and Eurus by nearly 40 points in chat score). On the other hand, Qwen RM is used roughly half the time for user prompts involving math, while Olmo and Eurus are used sparingly. This is consistent with Qwen RM's ranking on the "reasoning" split of

Figure 6: Utilization (%) of each RM on instruction-following queries from WildChat. The bars are arranged based on their overall scores on RewardBench, from lowest to highest. LASER dynamically selects from different RMs depending on the nature of the underlying instance.

RewardBench, outperforming Eurus and Olmo RMs by 15-20 points. Note that LASER *automatically* deduces these relative rankings of RMs and uses them depending on the underlying query *without having access to the RewardBench leaderboard*. Therefore, RM utilization of LASER can serve as an analysis tool for future work when assessing performance on untested domains.

**Generalization ability of LASER.** While recent work focuses on building RMs that reflect preferences across domains, a large body of prior work developed a suite of evaluation metrics catered to specific domains such as reasoning (Golovneva et al., 2022; Prasad et al., 2023). In Table 5 (Appendix C), we show that Llama-3-8B trained using LASER, to adaptively select relevant evaluation metrics, outperforms baselines by $1.62\%$ on average and can effectively filter underperforming metrics without degrading performance (c.f. Fig. 8). Furthermore, from Table 6, we observe that on reasoning datasets LASER outperforms sequential RM optimization under four different choices of loss functions: NLL, DPO (Rafailov et al., 2024b), DPO + NLL (Pang et al., 2024), and KTO (Ethayarajh et al., 2024). Finally, in Table 7, we find that models trained with LASER also exhibit the highest generalization to out-of-distribution settings such as on CommonsenseQA (Talmor et al., 2018) and MATH (Hendrycks et al., 2021c), reiterating the broad generalizability of LASER.

## 6    Conclusion

We present LASER, an adaptive method for selecting RMs and iteratively training LLMs using multiple RMs. We formulate the problem as a contextual multi-armed bandit problem, learning to select the RM that most improves the LLM conditioned on the given input or query. We test LASER across diverse settings, showing its utility on reasoning tasks, instruction-following tasks, and long-context generation. Across domains, we show that LASER *consistently* results in superior performance, whereas multi-RM baselines that select RMs using random or fixed strategies or ensemble multiple RMs uniformly have lower and more variable performance. In our analysis, we show that LASER is robust to noisy RMs, and flexibly uses different RMs depending on the domain, and generalizes to multiple settings. Lastly, by selecting one RM at a time, LASER provides the best of both worlds: consistently outperforming all baselines while still maintaining efficiency by only optimizing for one model at a time.

ETHICS STATEMENT

LLMs have been shown to reflect stereotypes, biases, and other negative traits contained in their pre-training data (Weidinger et al., 2021). Consequently, finetuned LLMs (including those trained with LASER) may also exhibit such undesirable traits in their generations during inference or training and exhibit the same potential for misuse as any other finetuned model. While prior work have made some headway in detecting such harmful content generated by LLMs (Inan et al., 2023), considerable research effort is needed in mitigating bias in LLMs. Conceptually, classifiers that detect risky, harmful, or biased content in the text can also be used as an additional RM in LASER's training to reinforce avoiding bias via preference optimization. However, we do not study this in our work and leave it to future work to explore these directions.

REPRODUCIBILITY STATEMENT

We provide comprehensive descriptions of our experimental setup, including the datasets, models, hyperparameters in Appendix A.1 and prompts for each dataset in Appendix E used across all experiments. The code for training and evaluation is included in `https://anonymous.4open.science/r/LASeR-5454/`. Furthermore, all pre-trained RMs and datasets used in this work are publicly available (link: `MMLU`, `MATH`, `GSM8K`, `StrategyQA`, `CommonsenseQA`, `WildChat`, `LongBench`).

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

# A EXPERIMENTS

## A.1 EXPERIMENTAL SETTING

**Training setup.** For training with LoRA, we set the rank to 16 and alpha to 32. We fine-tune the model for 10 iterations using a learning rate of $5e-6$ and a batch size of 16. Following Pang et al. (2024), we generate $P = 10$ pairs per problem for training with our loss in Sec. 3.1. For all experiments, we trained each method to converge. The number of iterations for each method was selected based on the observed convergence, with a performance metric threshold of $0.1$ across training batches on the dev set. The LinUCB algorithm has a total of 1.6M learnable parameters (including matrix $A$ and bias vector $b$). Our experiments are run on 4 RTX A6000 with 48G memory each.

**RewardBench.** Following Lambert et al. (2024), rewards are computed with no reference model and only use the log-likelihood of the reward model. For instance, given a reward model $\pi_{R^\star}$, the reward for an input $x_i$ and response $y_i$ is calculated as: $\log \pi_{R^\star}(y_i \mid x_i)$. There is no need for normalization since we use this log-likelihood to rank the responses. Specifically, on GSM8K, we computed the average and standard deviation of the rewards for the chosen responses as $-4.2965$ and $1.4156$, respectively, and for the rejected responses as $-5.9861$ and $1.6546$, respectively.

**Details of RMs.** We provide details for each chosen RMs:

- Zephyr-7B-Alpha: is a fine-tuned version of Mistral-7B model that was trained on on Ultra-Chat (Ding et al., 2023) and UltraFeedback (Cui et al., 2023) using DPO.
- Qwen1.5-7B-Chat: is pretrained with human-style conversation data inspired by Ouyang et al. (2022) along with questions, instructions, and answers in natural language, and post-trained with both SFT and DPO using diverse prompts (Lu et al., 2023).
- Eurus-7B-KTO: is a fine-tuned version of Eurus-7B-SFT model using KTO loss on UltraInteract (Yuan et al., 2024a) and UltraFeedback (Cui et al., 2023).
- OLMo-7B-Instruct: is the instruct version of OLMo-7B base model and was fine-tuned using UltraFeedback (Cui et al., 2023).

**Extracting embeddings for a query using $\pi_m$.** To extract embeddings for a query using $\pi_m$, we first process the input query through the policy model $\pi_m$. We use the embedding of the last token in

the query as the representation for the query. The embedding is then used as input to the subsequent bandit algorithm.

**Baselines.** Here we provide more details for baselines:

- **Classifier Selection.** We add an additional baseline that uses the RewardBench data to train a classifier that maps queries to an RM $\mathcal{C} : \mathbb{R}^d \to \mathcal{R}$, where $\mathcal{R} = R_1, R_2, \ldots, R_K$ is the set of RMs. Specifically, to construct a dataset for training $\mathcal{C}$, we take each query in the RewardBench data along with its corresponding chosen and rejected responses. The RewardBench dataset contains a total of 2985 examples across several categories including chat, safety, and reasoning. The dataset is split into a 80/20 ratio for training/development sets, then the classifier is trained on the training set and and validated on the development set. We use each RM to score these responses. The RM that assigns the correct score with the highest difference between the chosen and rejected response is selected to label the RM for that query. After training $\mathcal{C}$, we use this classifier to select the RM used for training the LLM in our pipeline. In the experiments, we use a three-layer MLP with hidden dimensions of 2048 and 1024 and an output dimension of 4 (number of RMs), with ReLU activation in each layer.
- **RM Ensembles.** While the ensemble methods generate scores from multiple RMs in a single iteration for a fixed set of responses sampled at the start of the iteration, we still generate new responses at each training iteration as part of the overall learning process. This ensures that the training dynamically incorporates updated responses from the LLM.

**Datasets.** For StrategyQA, GSM8K, and MMLU, we divided each dataset into training and test sets. The model is fine-tuned on the training set and dev set, then evaluate on the test set. For WildChat, the dataset was split into a 70/10/20 ratio for training, development, and testing. Following Zhao et al. (2024), prompt categorization is done using a small off-the-shelf classifier.[5] For LongBench, we subsample 5K examples for three tasks: multi-document QA, summarization, and few-shot learning. Each category was split into a 70/10/20 ratio, and the bandit model was trained and validated on the training and development sets and then tested on the test set. We report the detailed number of instances for train, development, and test sets in Table 3.

Table 3: Number of examples in train, val, and test sets.

| Task | Dataset/Category | Train | Dev | Test | Total |
|------|------------------|-------|-----|------|-------|
| Reasoning | StrategyQA | 1946 | 278 | 556 | 2780 |
| | GSM8K | 6750 | 750 | 1000 | 8500 |
| | MMLU | 11135 | 1591 | 3182 | 15908 |
| WildChat | Creative | 3500 | 500 | 1000 | 5000 |
| | Analysis | 3500 | 500 | 1000 | 5000 |
| | Coding | 3500 | 500 | 1000 | 5000 |
| | Factual | 3500 | 500 | 1000 | 5000 |
| | Math | 3500 | 500 | 1000 | 5000 |
| LongBench | Single-doc QA | 3534 | 505 | 1010 | 5049 |
| | Multi-doc QA | 3500 | 500 | 1000 | 5000 |
| | Summarization | 3500 | 500 | 1000 | 5000 |
| | Few-shot learning | 3500 | 500 | 1000 | 5000 |

A.2  DETAILS OF BANDIT ALGORITHMS

**Algorithm for Sec. 3.3.** We provide the detailed algorithm for Sec. 3.3 in Algorithm 1.

---

**Algorithm 1** Bandit-based Reward Model Selection for LLM Training

1: **Input:** LLM $\mathcal{M}$, reward models $\mathcal{R} = \{R_1, R_2, \ldots, R_K\}$, dataset $\mathcal{D} = \{x_1, x_2, \ldots, x_N\}$, bandit algorithm (LinUCB)
2: **Initialize:** Bandit algorithm parameters (e.g., $\theta_k$ for each RM)
3: **for** each training iteration $m = 1, 2, \ldots, M$ **do**
4:     **for** each batch or train step $t = 1, 2, \ldots, T$ **do**
5:         Select reward model $R_t^\star$ for time step $t$ using equation (2) (LinUCB)
6:         Sample a batch of samples from $\mathcal{D}$ and generate preference pairs following 3.1
7:         Fine-tune $\pi_m$ using preference pairs in $\mathcal{D}_{\text{pref}}$ using $\mathcal{L}^m$
8:         Update bandit parameters based on equation (3) (LinUCB)
9:     **end for**
10: **end for**

---

[5]Link: https://huggingface.co/valpy/prompt-classification

**Exp3.** Exp3 is a non-contextual bandit algorithm designed for adversarial settings. It maintains a probability distribution over the arms and selects arms based on the exponential weighting of past rewards. The probability for choosing arm $a_k$ at round $t$ is calculated as follows:

$$p_k(t) = (1 - \gamma) \frac{\exp(S_k(t))}{\sum_{a_k \in \mathcal{A}} \exp(S_k(t))} + \frac{\gamma}{K},$$

where $S_k(t)$ is the cumulative score for arm $a$ up to time $t$ and $\gamma$ is a parameter controlling the exploration rate.

The arm $a_k$ is selected by sampling the following categorical distribution

$$a_t \sim \text{Categorical}(p_1(t), \ldots, p_K(t)) \tag{4}$$

The score for arm $a_t$ is updated based on the observed normalized reward $-\hat{\mathcal{L}}^m(t)$ and the probability $p_k(t)$ of selecting that arm:

$$S_k(t + 1) = S_k(t) - \frac{\hat{\mathcal{L}}^m(t)}{p_k(t)} \cdot \mathbb{K}(a_t = a_k), \tag{5}$$

where $\mathbb{K}(a_t = a_k)$ is an indicator function that equals 1 if arm $a_k$ was selected at time $t$, and 0 otherwise.

**MAB reward normalization.** To maintain a consistent scale and magnitude MAB rewards across training, we apply scaled rewards based on the quantiles of the reward history, following the method outlined by Graves et al. (2017). Let $L = \{-\mathcal{L}^m(1), \ldots, -\mathcal{L}^m(t-1)\}$ represent the unscaled reward history up to time step $t$. This history's lower and upper quantiles are denoted as $q_t^{lo}$ and $q_t^{hi}$, respectively. We set $q_t^{lo}$ and $q_t^{hi}$ to be $20^{th}$ and $80^{th}$ quantiles. The scaled reward, $-\hat{\mathcal{L}}^m(t)$, becomes:

$$-\hat{\mathcal{L}}^m(t) = \begin{cases} 0 & \text{if} -\mathcal{L}^m(t) < q_t^{\text{lo}} \\ 1 & \text{if} -\mathcal{L}^m(t) > q_t^{\text{hi}} \\ \frac{-\mathcal{L}^m(t) - q_t^{\text{lo}}}{q_t^{\text{hi}} - q_t^{\text{lo}}} & \text{otherwise.} \end{cases}$$

## B  ADDITIONAL EMPIRICAL RESULTS

**LASER adaptively selects from multiple RMs on LongBench.** On LongBench (Fig. 7), we observe distinct utilization patterns for the QA tasks vs. summarization and few-shot learning. QA tasks exhibit nearly equal utilization of the top-2 RMs on RewardBench (Zephyr-7B-Alpha and Qwen1.5-7B-Chat in decreasing order), with the utilization of the Qwen RM even *exceeds* that of Zephyr RM for multi-document QA. In contrast, on summarization and few-shot learning the top RM (Zephyr) is far more preferred by LASER with margins of $59\%$ and $31\%$ over the second-best RM and the least performant RM being utilized less that $3\%$ of the times.

**Detailed results for each RM.** Here, we provide detailed reasoning results for each chosen RM where we use a single RM during training (c.f. Sec. 3.1) in Table 4. These results demonstrate that Qwen1.5-7B-Chat outperforms other RMs on StrategyQA and MMLU, whereas on GSM8K Zephyr-7b-alpha has the best performance with Llama-3-8B. However, LASER still yields the best performance, outperforming all RMs by at least $1\%$ on average across reasoning tasks, without the knowledge of which RM is most suited for each task a priori.

## C  GENERALIZATION CAPABILITIES OF LASER

**LASER Training with Domain-specific Evaluation Metrics.** While recent works focus on building RMs that reflect preferences across domains, an extensive body of prior work develops a suite of evaluation metrics catered to specific domains such as reasoning (Golovneva et al., 2022; Prasad et al., 2023). To show that LASER can be used to select any kind of evaluation metric from a collection of metrics during training, in Table 5, we present results with training LLMs on model-based metrics from ROSCOE (Golovneva et al., 2022) by replacing RMs with informativeness, faithfulness, reasoning alignment, hallucination, common sense error, semantic, coherence and perplexity

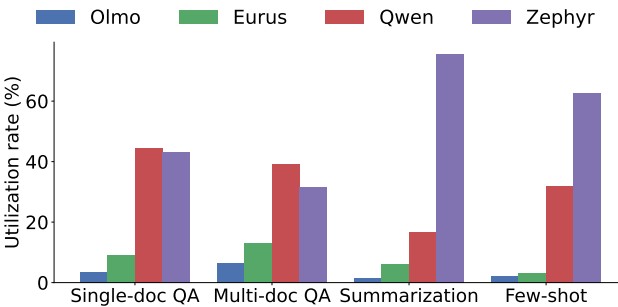

Figure 7: Utilization (%) of each RM on long-context understanding tasks. The bars are arranged based on their overall scores on LongBench, from lowest to highest. LASER dynamically selects from different RMs depending on the nature of the underlying instance.

Table 4: Performance of 4 RMs including OLMo, Eurus, Zephyr and Qwen1.5. Best is bolded, second-best is underlined.

| Method | Llama-3-8B | | | | Mistral-7B | | | |
|---|---|---|---|---|---|---|---|---|
| | StrategyQA | GSM8K | MMLU | Avg. | StrategyQA | GSM8K | MMLU | Avg. |
| OLMo-7B-Instruct | 80.23 | 68.91 | 65.74 | 71.62 | 68.73 | 44.96 | 56.94 | 56.88 |
| Eurus-7b-kto | 81.15 | 71.13 | 66.26 | 72.84 | 68.64 | 45.37 | 56.96 | 56.99 |
| Zephyr-7b-alpha | 84.29 | 73.16 | 67.15 | 74.87 | 70.06 | 45.81 | 62.04 | 59.30 |
| Qwen1.5-7B-Chat | 84.79 | 73.07 | 67.53 | 75.13 | 71.05 | 45.74 | 62.18 | 59.66 |
| LASER (Ours) | **85.96** | **74.75** | **68.24** | **76.32** | **73.06** | **46.89** | **62.27** | **60.74** |

Table 5: Comparison of LASER and baselines on ROSCOE. The baselines include supervised fine-tuning (SFT), sequential optimization, uniform rewards selection, and base model optimized with one specific evaluation metric (Perplexity, Informativeness).

| Method | Llama-3-8B | | | | Mistral-7B | | | |
|---|---|---|---|---|---|---|---|---|
| | StrategyQA | GSM8K | MMLU | Avg. | StrategyQA | GSM8K | MMLU | Avg. |
| SFT | 80.41 | 69.43 | 65.66 | 71.83 | 68.57 | 43.62 | 56.48 | 56.22 |
| Perplexity | 80.55 | 69.21 | 65.62 | 71.79 | 68.83 | 43.47 | 57.14 | 56.48 |
| Informativeness | 82.87 | 73.55 | 66.69 | 74.37 | 70.29 | **44.98** | 59.29 | 58.19 |
| Random RM Selection | 82.72 | 70.93 | 66.10 | 73.25 | 69.24 | 44.05 | 57.68 | 56.99 |
| Sequential RM Selection | 83.15 | 73.38 | 66.17 | 74.23 | 70.40 | 44.79 | 59.07 | 58.09 |
| LASER (Ours) | **83.54** | **73.80** | **66.79** | **74.71** | **70.91** | 44.93 | **59.63** | **58.49** |

in Sec. 3. Llama-3-8B models trained using LASER yield 1.62% accuracy improvement over baselines on average across three datasets. These results are also generalized to Mistral-7B, except for GSM8K, where we achieve comparable performance to the Base + Informativeness baselines. Note that the perplexity of most responses is nearly identical, making it difficult to distinguish between them, explaining why perplexity shows little to no improvement compared to the base model.

**Robustness to Underperforming Evaluation Metrics.** Similar to our analysis on noise in rewards in Sec. 5, we investigate how adding ROSCOE metrics with poor correlation to human-annotated labels in meta-evaluation by Golovneva et al. (2022) impacts the performance of Llama-3-8B on GSM8K. Once again, even with ROSCOE metrics, demonstrates LASER can maintain consistent performance by adaptively prioritizing

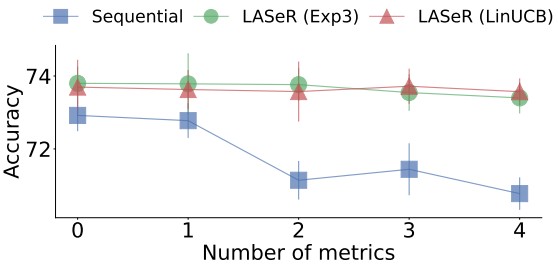

Figure 8: Impact of irrelevant metrics from ROSCOE on the GSM8K accuracy of LASER and sequential baseline.

the most relevant reward signals, outperforming the sequential baseline, which fails to filter out irrelevant information effectively. Fig. 8 shows that as the number of irrelevant metrics increases, LASER's selection strategy continues to perform robustly. Specifically, LASER has an average performance drop of only 0.13%, whereas the sequential baseline suffers a 2.15% accuracy drop

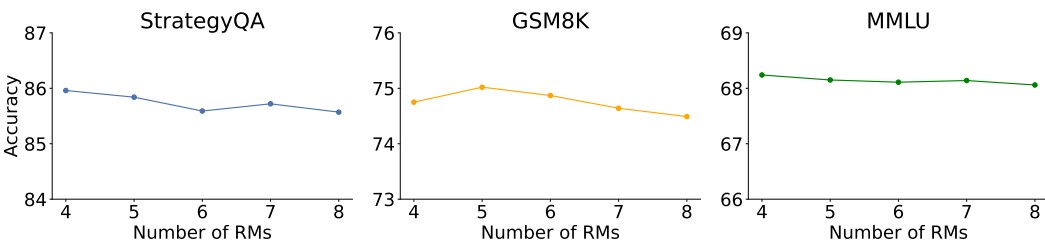

Figure 9: LASER's performance is robust to adding weaker RMs to the candidate set to select from.

under the same conditions. Lastly, LASER using Exp3 maintains a consistent performance level with a $0.4\%$ accuracy drop.

**Generalization to Training Loss Functions.** In Sec. 3.1, we state that the choice of loss function used to train the LLM depends on the underlying task or domain. Nevertheless, we always use the training loss as the MAB reward to update the MAB's parameters. Here we study the performance of LASER and baselines with 4 different loss functions, NLL, DPO, NLL + DPO (Pang et al., 2024), and KTO (Ethayarajh et al., 2024), in the reasoning domain. Results in Table 6 show that training LLMs with multiple rewards using LASER outperforms sequential RM

Table 6: Across different training loss functions, optimizing with multiple RMs via LASER outperforms the sequential RM selection with Llama-3-8B. SQA denotes StrategyQA.

| Loss | Method | SQA | GSM8K | MMLU | Avg. |
|---|---|---|---|---|---|
| NLL | Sequential | 82.75 | 71.80 | 65.41 | 73.32 |
| | LASER | 85.11 | 74.94 | 67.09 | **75.71** |
| DPO | Sequential | 83.26 | 71.94 | 65.38 | 73.53 |
| | LASER | 84.71 | 73.94 | 67.02 | **75.22** |
| KTO | Sequential | 83.62 | 73.07 | 69.02 | 75.24 |
| | LASER | 84.87 | 73.86 | 69.05 | **75.66** |
| NLL+DPO | Sequential | 83.90 | 72.94 | 68.02 | 74.95 |
| | LASER w/. Acc | 83.04 | 73.12 | 65.46 | 73.87 |
| | LASER | 85.96 | 74.75 | 68.24 | **76.24** |

selection by $2.4\%$, $1.7\%$, and $1.3\%$ when using NLL, DPO, NLL+DPO loss functions, respectively; while both methods yield comparable performance with KTO. Additionally, we found that the most effective training loss functions are NLL + DPO for StrategyQA, NLL for GSM8K, and KTO for MMLU. However, irrespective of the choice of the underlying loss function, LASER is more effective at balancing and adaptively selecting from multiple RMs. Lastly, we compare LASER with a variant in which we use $\text{Acc}(y^w) - \text{Acc}(y^l)$ as the MAB reward, which uses the ground-truth information about the final answer. We find that using the negative training loss of the LLM is more effective than using accuracy as the MAB reward.

**Generalization to OOD Tasks.** We first assess the generalization ability of our method by training models on specific datasets and evaluating their performance on out-of-distribution reasoning tasks. Specifically, we train the model on the StrategyQA and MMLU datasets and evaluate its generalization on the CommonsenseQA (CSQA; Talmor et al., 2019) dataset. Similarly, we train on GSM8K and test on MATH (Hendrycks et al., 2021c) to assess the model's ability to generalize across different reasoning datasets. These tasks are designed to capture both general reasoning ability and OOD generalization across domains. We report the results in Table 7, where we find that across both Llama-3-8B and Mistral-7B models, models trained with LASER yield the best average

Table 7: Generalization performance of different models trained on StrategyQA, MMLU, and GSM8K, and evaluated on CSQA and MATH, respectively.

| Method | Llama-3-8B | | | Mistral-7B | | |
|---|---|---|---|---|---|---|
| | CSQA | MATH | Avg. | CSQA | MATH | Avg. |
| SFT | 65.64 | 29.13 | 47.39 | 59.06 | 16.38 | 37.72 |
| Best RM | 67.59 | 31.54 | 49.57 | 60.46 | 18.08 | 39.27 |
| Avg. RM | 67.16 | 30.36 | 48.76 | 60.06 | 17.25 | 38.66 |
| Random RM Selection | 68.31 | 30.21 | 49.26 | 60.19 | 16.96 | 38.58 |
| Sequential RM Selection | 67.73 | 30.25 | 48.99 | 60.56 | 17.96 | 39.26 |
| LASER (Ours) | **69.26** | **31.67** | **50.47** | **61.65** | **18.97** | **40.31** |

accuracy beating training with the best RM by roughly $2\%$ (absolute) on CSQA with Llama-3-8B. On Mistral-7B, training with LASER outperforms both training with single best RM and sequential RM selection by slightly over $1\%$.

**Generalization to the Number of RMs.** To study the generalization capability of LASER across the number of RMs, we expand the candidate set of RMs with up to 4 more RMs from the Reward-Bench leaderboard, including Tulu-2-DPO-7B (Ivison et al., 2023), Zephyr-7B-Gemma (Tunstall & Schmid, 2024), Qwen1.5-MoE-A2.7B-Chat (Team, 2024), Archangel-7B (Ethayarajh et al., 2024). Fig. 9 shows that the accuracy remains consistent across all datasets as the number of RMs varies. StrategyQA remains near 85.9%, GSM8K around 74.8%, and MMLU close to 68.1%, with minimal fluctuations, indicating robust performance regardless of the number of RMs.

## D    ADDITIONAL DISCUSSION

**Related Work: Multi-Armed Bandits (MABs).** There is a long-standing history of using multi-armed bandit algorithms for diverse applications in machine learning spanning online advertising (Chen et al., 2013), recommendation systems (Li et al., 2010), hyperparameter optimization (Li et al., 2018), curriculum learning (Graves et al., 2017), with some recent work at the intersection of MABs and language models (Pasunuru et al., 2020; Krishnamurthy et al., 2024; Dwaracherla et al., 2024, *inter alia*). In the realm of RLHF specifically, Dwaracherla et al. (2024) use double Thomson Sampling (Wu & Liu, 2016) to select which of the sampled responses should be annotated and paired using a single fixed RM, improving LLM performance. In contrast, LASER first selects which RM (i.e., model of preferences) should be used to annotate LLM's responses to a query and then creates multiple preference pairs from these responses. Pasunuru et al. (2020) optimize text generation models for different evaluation metrics such as ROUGE-L and BLEU via policy gradients (Williams, 1992) over existing (static) question and data-to-text generation datasets. In contrast, LASER adopts an iterative training recipe and improves downstream generation performance across a wide range of tasks from instruction-following to math and commonsense reasoning by selecting relevant RMs and scoring the LLM's own generated responses without access to true rewards, i.e., gold labels, and without optimizing for the downstream evaluation metric.

**LASER with different "kinds" of RMs.** In Sec. 4.2, we show that LASER can choose from a set of candidate RMs, and our analysis in Fig. 3 highlights the fact that LASER is robust to noisy RMs. In Appendix C, we show that LASER can also be used with metric-based rewards (Golovneva et al., 2022). These experiments reflect a conceptual split between the generator (the LLM) and the scorer (the RM or metric). Thus, LASER is applicable to other settings that follow this paradigm, e.g., using an LLM-as-a-judge (Zheng et al., 2023), where LASER could be used to choose between different judge models, prompts, or different combinations of RMs and metrics. However, consistent with the "self-preference" bias of LLMs (Panickssery et al., 2024), we caution that using an RM that is based on the same model as the LLM used for generating responses could lead to the MAB spuriously favoring certain RMs. We leave further study on extending LASER to future work.

**Quality of RMs used with LASER.** Methods that rely on RMs for scoring generally assume that these RMs have a strong correlation with human judgments. LASER tempers this assumption in a number of ways: first, by ensembling multiple RMs, LASER weakens the effect of noisy RMs; this can be seen in Fig. 3, where LASER mitigates the negative impact of a noisy RM even as the level of noise is increased. Moreover, the fact that LASER can select RMs at an instance-level means that there need not be a *single* RM that always correlates well across all instances. However, LASER does require at least one RM to be positively correlated with human judgments on each instance. If this assumption is not met (i.e., *all* RMs are poorly-correlated across *all* instances), then optimizing for the RMs will yield poor results. Note that this holds true for any method optimizing for RMs. Because LASER selects from multiple RMs, its contributions are complementary to developments in RMs, which can easily be integrated into LASER, as well as improvements to preference optimization loss functions (see Appendix C). Such RM improvements are likely to be necessary as LLMs are deployed in domains that are out of scope for existing systems and domains with heterogeneous requirements (e.g., our generation domains in Sec. 4.2). In these cases, there will be no single "perfect" existing RM, and successful solutions will likely involve mixing multiple RMs. A core benefit of LASER is its the ability to automatically filter RMs; in Fig. 6 we see that utilization differs across domains. This allows users to avoid expensive experimentation with subsets of RMs: they can simply offload this task to LASER, which will automatically select the more useful RM(s).

# E PROMPTS

---

**Reasoning**

**Prompt:** Your task is to answer the question below. Give step by step reasoning before you answer, and when you're ready to answer, please use the format "Final answer:..."
**Question:** {input}
**Solution:**

---

**Long-Context Understanding**

**Single-Doc QA:**
**Prompt:** You are given a scientific article and a question. Answer the question as concisely as you can, using a single phrase or sentence if possible. If the question cannot be answered based on the information in the article, write "unanswerable". If the question is a yes/no question, answer "yes", "no", or "unanswerable". Do not provide any explanation.
Article: context Answer the question based on the above article as concisely as you can, using a single phrase or sentence if possible. If the question cannot be answered based on the information in the article, write "unanswerable". If the question is a yes/no question, answer "yes", "no", or "unanswerable". Do not provide any explanation.
**Question:** {input}
**Answer:**

**Multi-Doc QA:**
**Prompt:** Answer the question based on the given passages. Only give me the answer and do not output any other words. The following are given passages.
{context}
Answer the question based on the given passages. Only give me the answer and do not output any other words.
**Question:** {input}
**Answer:**

**Summarization:**
**Prompt:** You are given several news passages. Write a one-page summary of all news.
News: {context}
Now, write a one-page summary of all the news.
**Summary:**

**Few-shot Learning:**
**Prompt:** Answer the question based on the given passage. Only give me the answer and do not output any other words. The following are some examples.
{context}
**Question:** {input}
**Answer:**

---

**Instruction-Following**

**Prompt:** You are an assistant capable of assisting users in various tasks, including creative writing, analysis of texts and data, coding, providing factual information, and solving math problems. For creative writing, help users brainstorm ideas and develop their narratives. For analysis, guide users in breaking down concepts and exploring different perspectives. In coding, assist with programming questions and debugging. When providing factual information, ensure accuracy and cite reliable sources. For math reasoning, offer step-by-step solutions and encourage logical thinking. Maintain a clear, engaging, and supportive tone throughout your responses to foster learning and creativity.
**Question**: {input}
**Answer:**

