# OpenReview forum: "LASeR: Learning to Adaptively Select Reward Models with Multi-Armed Bandits"
_ICLR.cc/2025/Conference — Submitted to ICLR 2025_

### Official Review · Reviewer_kZ21 · 2024-10-20

**Soundness:** 3
**Presentation:** 3
**Contribution:** 3
**Rating:** 8
**Confidence:** 3

**Summary:**

This paper describes LASER, which:
- Uses an ensemble of reward models (specifically 4) to align language models
- Selects reward models with a multi-armed bandit (based on the LinUCB algorithm) to classify the (embedded) input into different RM tasks, and
- Uses the chosen RM with language model sampling to create and train on preference data.

**Strengths:**

- The paper is well written
- Using LinUCB is well motivated
- Solution makes intuitive sense
- Experiments are extensive

**Weaknesses:**

Nitpicky things:
- (As far as I know) Figure 2 is not referred to in the main text
- Add a y-axis to Figure 4
- (As far as I know) Appendix B is not referred to in the main text
- I could not find where it mentions how many preference pairs are generated per iteration/

**Questions:**

I have two questions:
1. Why does the ensemble baseline perform worse than the best RM? This is indicated in lines 95-97 and Table 1.
2. You mention that Table 3 shows results on long-context understanding without training and best-of-n sampling. Do you have any results where you use the RM to generate preference pairs and use those as in-context examples?

---

> ### Author Response · Authors · 2024-11-19
> **Response to Reviewer kZ21**
>
> We thank you for your review and detailed suggestions.  We appreciate you recognizing our work as “well-written” and “well-motivated” and highlighting the fact that our “solution makes intuitive sense”, as well as our “extensive” experiments. We respond to your suggestions and comments in detail below and have revised our paper to reflect these points:
>
> > (As far as I know) Figure 2, Figure 4 and Appendix B
>
> Thanks for your comment. We have revised the paper according to your suggestion.
>
> > I could not find where it mentions how many preference pairs are generated per iteration
>
> As we clarify in footnote 1, for each query x in each iteration, we generate 10 corresponding preference pairs. The total number of queries in the training set varies with each dataset and is listed in Table 3 of Appendix A.1.
>
> > Why does the ensemble baseline perform worse than the best RM? This is indicated in lines 95-97 and Table 1.
>
> The RM Ensemble combines the output scores of multiple RMs, some of which may be noisy (used in an out-of-distribution setting) and provide conflicting signals (see Fig 5., Sec 5. lines 473-495 in the updated PDF), overall making it less effective for training. In contrast, the Best RM selection relies on a single RM at a time, eliminating scaling issues and conflicts and resulting in a cleaner and more reliable training signal. Additionally, the Best RM is chosen based on the highest overall score on RewardBench, indicating its ability to perform well across various tasks and domains. We have added this clarification in lines 492-495 of Sec. 5 in our revised paper.
>
> > You mention that Table 3 shows results on long-context understanding without training and best-of-n sampling. Do you have any results where you use the RM to generate preference pairs and use those as in-context examples?
>
> Thank you for the suggestion. To the best of our knowledge, using in-context examples for alignment is typically employed for tasks with shorter queries as done in [1] and not for long-context tasks like the ones in our experiments. When working with longer inputs, LLMs already struggle with position bias and focusing on information provided in the input (as noted in [2]). Therefore, we expect in-context examples to also have a distracting effect, and do not experiment with it further, leaving such a study to future work.
>
> [1] Lin, Bill Yuchen, et al. "The unlocking spell on base LLMs: Rethinking alignment via in-context learning." ICLR 2023. https://arxiv.org/abs/2312.01552
>
> [2] Liu, Nelson F., et al. "Lost in the middle: How language models use long contexts." TACL 2023. https://arxiv.org/abs/2307.03172

---

> ### Comment · Reviewer_kZ21 · 2024-11-19
>
> Thank you for the clarifications, I have increased my score - good luck!

---

> > ### Author Response · Authors · 2024-11-19
> > **Thank you**
> >
> > Thank you for acknowledging our rebuttal and for raising the score. We appreciate your time and effort in reviewing our paper.

---

### Official Review · Reviewer_eLcv · 2024-11-03

**Soundness:** 2
**Presentation:** 2
**Contribution:** 2
**Rating:** 5
**Confidence:** 3

**Summary:**

The paper introduce the LASER framework, which enables LLMs to adaptively select the most appropriate Reward Models for different tasks. This approach addresses the challenges of using multiple RMs by optimizing the selection process, combining with bandit algorithm, such as LinUCB and Exp3, thereby enhancing the performance of LLMs across reasoning, QA and instruction following tasks.

**Strengths:**

1. Innovative approach to automatically selecting Reward Models using bandit algorithm. This MAB framework, allowing the model to dynamically choose the most suitable RM at instance level based on contextual information (embedding of the last token). This avoid the computational burden of previous approaches that based on RM ensemble.

2. Empirical results presented in the paper indicate that LASER achieves performance improvements over traditional methods, such as ensemble RM scores. LASER achieve improvement in wide range of tasks, suggest it's generalibility.

**Weaknesses:**

A notable weakness of the paper is the approach taken to set the reward in the MAB problem as the negative training loss. While maximizing the reward is equivalent to minimizing the training loss, this method raises concerns about the alignment of selected preference pairs with the language model. Specifically, if the MAB algorithm consistently selects RM that align with the LM (means that the preference pair generate by the RM has low loss in LM), it's essentially just reinforce the LM's existing biases rather than addressing its mistakes.

This raises a critical question, why would selecting RM that align with the LM lead to improved performance? If the LM makes an error and the RM correctly identifies that mistake, the training loss would be high, which means the MAB algorithm might avoid selecting the RM that highlights the LM's shortcomings. Consequently, this could result in a scenario where the model becomes less capable of correcting its errors, as it may favor RMs that validate its outputs rather than challenge them.

**Questions:**

Can author address why chosing the reward as the negative train loss would help in mitigating bias of LLM?

---

> ### Author Response · Authors · 2024-11-19
> **Response to Reviewer eLcv**
>
> Thank you for your detailed review and for highlighting our “innovative approach” and “performance improvements”.  We would like to clarify your main question on why we choose the negative loss as the MAB reward for RM selection. While it is possible that in some cases, the selected RM could reinforce LLM biases, we believe that it does not occur in our current experimental setup for the following reasons:
>
> * **Downstream Performance:** Empirically, if the chosen RMs were only superficially agreeing with the LLM while ignoring its mistakes, the model trained with these faulty preferences would not improve downstream performance. However, this is not what we observe in Sec 4.2, with LASeR consistently outperforming the base model across multiple tasks and domains.
> * **Selection of candidate RMs:** As we mention in lines 290-294 in the updated PDF, we choose the set of candidate RMs based on their strong performance on RewardBench leaderboard at 7B scale. By design, RewardBench (Lambert et al. 2024) rates RMs on their ability to align with human preferences for diverse domains. So, in order to get high RewardBench scores, the RMs (including our chosen ones) are incentivized to not overfit to one particular LM. As per your suggestion, we briefly discuss the risk of selected RMs reinforcing mistakes of LLMs in Appendix D of the revised draft.
> * **Case study on reasoning tasks:** Since LLM responses on reasoning tasks can be easily evaluated based on correctness of the final answer, as per your suggestion, we conduct a case study of using MAB reward as the Acc(chosen) - Acc(rejected) calculated w.r.t. the ground-truth answer while training the LLM with DPO + NLL loss (discussed in Sec 3.1). This would avoid scenarios where the RM is selected solely based on agreement with the LLM (loss being already low).
>     * **Results:** The results are presented in Table 6 where we observe that given the same loss function for training the LLM loss, **setting the MAB reward as the negative loss outperforms the using accuracy as the MAB reward by 2.37% points averaged across the three reasoning tasks.** We believe that if the negative loss was skewing LASeR’s RM selection to spuriously agree with the LLM, then accuracy-based MAB rewards would have yielded better reasoning accuracy.
>
>  | Loss       | Method              | SQA    | GSM8K  | MMLU   | Avg.   |
> |------------|---------------------|--------|--------|--------|--------|
> | NLL+DPO    |  LASER w/. Acc | 83.04  | 73.12  | 65.46  | 73.87  |
> |            | LASER w/. Negative LLM loss               | 85.96 | 74.75 | 68.24 | **76.24** |
>
>
>
> We hope this clarifies your query about using the LLM’s loss as the MAB reward.

---

> > ### Author Response · Authors · 2024-11-23
> > **Follow-up reminder for Reviewer eLcv**
> >
> > Thank you once again for your valuable feedback. We hope our response has addressed all of your questions and will allow you to revisit your score, otherwise we would be happy to engage further and address any further questions you might have in the remaining few days of the discussion period.

---

> > > ### Author Response · Authors · 2024-11-24
> > > **Reminder to reviewer eLcv**
> > >
> > > Since the rebuttal period is drawing to a close, with only 2 days left before the 26th, we wanted to check in again and see whether our additional experiments/positive results and responses have addressed your comments and will allow you to revisit your score.

---

> > > > ### Author Response · Authors · 2024-11-27
> > > > **Reminder to reviewer eLcv**
> > > >
> > > > Since today is the final day for updating the PDF for our submission, we wanted to kindly check in again to see whether our rebuttal and revised paper have addressed all your concerns. If so, we would appreciate it if you could revisit your score. Otherwise, we are happy to continue discussing any remaining questions, since the rebuttal period has been extended.

---

> > > > > ### Author Response · Authors · 2024-11-29
> > > > > **Reminder to Reviewer eLcv**
> > > > >
> > > > > Dear Reviewer eLcv, we wanted to check in again to see whether our rebuttal and revised paper have addressed all your comments. If so, we would appreciate it if you could revisit your score. Otherwise, we are happy to continue discussing any remaining questions, since the rebuttal period has been extended.

---

> > > > > > ### Author Response · Authors · 2024-12-02
> > > > > > **Reminder to Reviewer eLcv**
> > > > > >
> > > > > > Since today is the deadline for the authors-reviewers discussion period, we wanted to kindly check in to see if our rebuttal and revised paper have addressed all your comments. If so, we would greatly appreciate it if you could revisit your score. Otherwise, we are happy to continue discussing any remaining questions until the end of the discussion period.

---

### Official Review · Reviewer_QEwv · 2024-11-04

**Soundness:** 2
**Presentation:** 2
**Contribution:** 3
**Rating:** 5
**Confidence:** 4

**Summary:**

This paper introduces LASeR, a novel approach for training large language models (LLMs) by adaptively selecting suitable reward models (RMs) for various tasks, framed as a multi-armed bandit problem. Unlike previous methods that ensemble multiple RMs, LASeR aims to enhance computational efficiency and avoids conflicts from using different RMs, by selecting and utilizing the most appropriate RM for each instance to rank outputs and generate preference data. Extensive experiments demonstrate that LASeR consistently outperforms baseline methods across multiple benchmarks.

**Strengths:**

1. The proposed LASER iteratively trains LLMs using different RMs by dynamically selecting the most appropriate one for each training instance using a contextual bandit algorithm, specifically LinUCB, which effectively addresses the potential inefficiencies and conflicts present in ensemble methods that handle multiple RMs simultaneously.
2. The paper thoroughly evaluates LASeR across various datasets and tasks, showcasing its superior performance over baselines.

**Weaknesses:**

1. Fairness of Comparison: In Section 4, LASeR is compared with baselines that do not actively leverage information from interactions with the datasets. The RM selection in these baselines is either fully random or in offline fashion. Specifically, in Table 4, the "best RM" baseline (Zephyr-7b-alpha) is not the top performer on StrategyQA and MMLU, which contradicts claims in Appendix B. Additionally, the sequential RM selection shows better performance than other baselines across most datasets, yet the explanation for this surprising result is missing. It would be more appropriate to compare LASeR with ensemble methods where RM aggregation adapts based on training information, as LASeR leverages training loss to adapt RM selection.
2. Limited Comparison with Ensemble Methods: The paper only compares LASeR with a simple offline RM Ensemble using averaged scores, despite the existence of more sophisticated ensemble methods that incorporate training signals [A1] [A2]. It remains unclear whether LASeR outperforms such methods. Moreover, the results on reasoning are the only ones compared against Offline RM Ensemble. Due to these issues, it is insufficient to infer whether LASeR outperforms ensemble methods in terms of robustness to noisy rewards or different tasks like long-context understanding and instruction-following tasks.
3. Clarification of Unique Contributions: While the paper effectively incorporates LinUCB for RM selection in LLM training, there is insufficient discussion on the challenges encountered during this integration. Highlighting the unique contributions and the advantages over ensemble methods would strengthen the paper.
4. Efficiency Concerns Beyond Accuracy-Time Tradeoff: The paper does not address that, in high-dimensional embedding spaces or with a large number of RMs, LASeR’s need to compute the inverse of the covariance matrix A for each arm could be computationally intensive.
5. Minor issues:
   1. Undefined/Misleading Notations: For example, in Equation (1), $x$ should be $x_i$, and $\sigma,\beta$ are not defined previously.  Footnote 2 mentions $P$ as the number of sampled pairs per prompt, while $n$ in the notation paragraph seems to have a similar purpose. In section 3.2, the notation should be $\mathbf{x}_{m,t}$ to represent the batch at the $t$-th step of the $m$-th iteration. Also, $\mathbf{x}_t$ should be defined clearly as the set of prompts, but is later used as the set of the $d$-dimensional embedding of the prompt. The left-hand side of equation (2) should be the index of the selected RM, while $R_t^{*}$ is the model itself.
   2. Grammar and Typos: Some sentences, such as those describing the stages of LASER, contain grammatical issues or are incomplete. The reviewer recommends a thorough review for consistency in symbols and text.


[A1] Zhang, Shun, et al. "Improving reinforcement learning from human feedback with efficient reward model ensemble." arXiv preprint arXiv:2401.16635 (2024).

[A2] Wang, Zihao, et al. "Transforming and Combining Rewards for Aligning Large Language Models." arXiv preprint arXiv:2402.00742 (2024).

**Questions:**

1. Could the authors elaborate the definition of $e_m$ and the method for generating the sentence embedding from the model $\pi_m$? Which part of Figure 1 corresponds to this process?
2. Could the authors provide insights into why Random RM selection often outperforms the Offline RM Ensemble and why Sequential RM selection surpasses Random RM selection?

---

> ### Author Response · Authors · 2024-11-19
> **Response to Reviewer QEwv [Part 1]**
>
> We thank you for your detailed review and for acknowledging our “novel” method LASeR for dynamically selecting RMs and training LLMs as well as our “thorough” evaluation demonstrating LASeR’s “superior performance over baselines”. Please find the responses to your comments below.
>
> > [W1] Fairness of Comparison
>
> We would like to clarify that Zephyr-7b-alpha is the RM with the *best overall score* on RewardBench, but it’s not necessarily best for every task. For example on RewardBench, Qwen has slightly higher reasoning scores than Zephyr. We add the best RM baseline because when we do not know which RM to use, then a natural approach is to pick the best overall score. For consistency we use Zephyr-7b-alpha as the best RM across all three domains: reasoning, instruction-following and long-context  (as mentioned in lines 314-317 in the updated pdf). Based on your comment, we add LASeR’s performance to the comparison of RMs in Table 4, which still shows that LASeR outperforms Qwen RM across models and reasoning tasks, without using a priori knowledge of which RM is best suited. We have updated the text in Appendix B to reflect these findings. As for your point about RM ensembles, please refer to the response below.
>
> > [W2] Limited Comparison with Ensemble Methods
>
> **Discussion of A1**: A1 trains many RMs that share a common base model and different heads (significantly smaller in size). Then for training LLM they have an ensemble score of RMs for PPO optimization. Since we do not control RM training in our work, each RM is a separate stand-alone 7B-scale model. While we aggregate the score of RMs to get a joint signal in the “RM Score Ensemble” baseline, this combination is not online as that would be computationally expensive. Instead, at the start of *each iteration*, we generate the responses from the current LLM (trained from the past iteration), score responses according to each RM, and then create preference pairs for training the model in the current iteration.
>
> **Discussion of A2**: This work combines preference signals from multiple RMs using a logical AND operation. Inspired by this, we implement a heuristic as follows: We sample 32 responses for each query from which we can create (32 choose 2) pairs. We randomly sample 100 pairs from this set and rate whether each RM assigns a higher preference to the first response: i.e., RM(Response 1) > RM(Response 2). In the end, we select 10 pairs that have the highest agreement in preference rankings among multiple RMs. In other words, we first form pairs where all 4 RMs agree over the chosen response, followed by agreement from 3 RMs, and so on. We report this as the “RM Agreement Ensemble” baseline in our revised paper, and find that while agreement-based ensembling is better than score based ensembling, **LASeR still consistently outperforms RM Agreement Ensemble baseline, e.g. by average of ~2% on reasoning tasks and 63.77% Alpaca win rate on WildChat with Mistral-7B.**
>
> Additionally, we **also add ensemble baselines for both WildChat and LongBench in Sec 4.2**, showcasing the superior performance of LASeR over RM ensembles across multiple domains.
>
> > [W3] MAB integration issues:
>
> Implementationally, we found that LinUCB was easy to integrate into LLM training. This was in part due to our approach of using iterative training with model-generated data, which provides an easily-accessible signal for updating the MAB. In terms of computational overhead, adding the MAB did not pose any problems, as its size is far smaller than the LLMs being trained: the MAB is an MLP with 1.6M parameters in the LinUCB setting and can be jointly trained along with the LLM (7B-8B parameters). We have uploaded our code along with the submission and will open-source it for the community to experiment with.
>
> > [W4]  Efficiency Concerns Beyond Accuracy-Time Tradeoff:
>
> Your point on covariance inverse computation is well-taken. Empirically, we find that for d=4096 and using 4 reward models, inverse computation needs to be performed for each RM once per batch (and can be cached to be reused later), which **only adds a latency of 1.27 seconds**. In comparison, a forward and backward pass of the LLM takes 36.39 seconds. More efficient approximations/inversion methods could be adopted here to further streamline LASeR’s training.

---

> > ### Author Response · Authors · 2024-11-19
> > **Response to Reviewer QEwv [Part 2]**
> >
> > > [W5] Minor issues:
> >
> > Thank you for pointing these out, we have modified the notations as per your suggestion in the updated version. We would also like to clarify that while we sample n=32 responses for each query, it can be used to create (32 choose 2) pairs, of which we randomly sample 10 for training the LLM.
> >
> > > [Q1] Could the authors elaborate the definition of em and the method for generating the sentence embedding from the model $\pi_m$? Which part of Figure 1 corresponds to this process?
> >
> > To extract embeddings for a query using $\pi_m$, we first process the input query through the policy model $\pi_m$. We use the embedding of the last token in the query as the representation for the query. The embedding is then used as input to the subsequent bandit algorithm. Thank you for raising this and we have added this clarification in Appendix A.1. We do not show the step converting each query into an embedding Fig. 2 for easier visualization, but we highlight it in Sec 3.2 and Appendix A.1 lines 863-866 (updated pdf).
> >
> > > [Q2] Could the authors provide insights into why Random RM selection often outperforms the Offline RM Ensemble and why Sequential RM selection surpasses Random RM selection?
> >
> > The RM Ensemble combines the output scores of multiple RMs, some of which may be noisy (used in an out-of-distribution setting) and provide conflicting signals (see Fig 5., Sec 5. Lines 473-495 in the updated pdf), overall making it less effective for training. In contrast, Random RM selection uses only one RM at a time, avoiding these conflicts. While Random RM selection may not always pick the optimal RM for a query, the absence of conflicting signals generally leads to a cleaner training signal compared to the noisy ensemble approach. We have added this clarification in lines 493-495 of Sec. 5 in our revised paper.
> >
> > Sequential RM selection systematically explores different RMs in a structured manner, ensuring that each RM contributes to the training process over time. This exploration reduces the risk of over-relying on poor-performing RMs for extended periods, providing a more balanced and diverse training signal. On the other hand, Random RM selection explores the RMs without such structure, which means it might choose multiple poorly performing RMs consecutively, leading to suboptimal training signals early on and potentially degrading overall performance.

---

> > > ### Author Response · Authors · 2024-11-23
> > > **Follow-up reminder for Reviewer QEwv**
> > >
> > > Thank you once again for your valuable feedback. We hope our response has addressed all of your questions and will allow you to revisit your score, otherwise we would be happy to engage further and address any further questions you might have in the remaining few days of the discussion period.

---

> > > > ### Author Response · Authors · 2024-11-24
> > > > **Reminder to Reviewer QEwv**
> > > >
> > > > Since the rebuttal period is drawing to a close, with only 2 days left before the 26th, we wanted to check in again and see whether our responses have addressed your comments and will allow you to revisit your score.

---

> > > > > ### Comment · Reviewer_QEwv · 2024-11-25
> > > > >
> > > > > Thank you for your efforts and the detailed clarification. After carefully reviewing the responses and the additional results, I think adaptive RM selection is an interesting direction. However, it appears that the current approach may not efficiently utilize the training signals. Comparing this with the added baselines (from Tables 1, 2, and 4) that do not require training signals, the improvement provided by LASeR, while valuable, seems relatively modest (i.e., less than 2%) and comes at the cost of increased computational resources, including additional memory and computation for using LinUCB in the RM selection learning process. Thus, I have decided to maintain my current score.

---

> > > > > > ### Author Response · Authors · 2024-11-26
> > > > > > **Follow-up Response to Reviewer QEwv**
> > > > > >
> > > > > > Thank you for your thoughtful feedback and for the engagement. We appreciate your acknowledgment that adaptive RM selection is an interesting direction and your recognition of the value provided by LASeR. We would like to address your remaining concerns as follows:
> > > > > >
> > > > > > **Performance Improvement**: We respectfully disagree with this observation and present the following arguments:
> > > > > > * We believe that LASeR consistently demonstrates superior performance across all datasets and models. In contrast, other methods exhibit inconsistent results. For example, on the StrategyQA dataset, Random RM Selection ranks as the second-best baseline with Llama-3-8B but performs significantly worse with Mistral-7B, demonstrating its lack of generalizability. Importantly, the gains achieved by LASeR are consistent across diverse settings.
> > > > > > * In the setting where adaptive RM selection is most conducive when training LLM queries from multiple diverse domains such as the instruction-following setting (lines 409-423). In a head-to-head comparison (including additional baselines) we show that **LASeR outperforms the baselines with a substantially higher AlpacaEval win rate** (see table below and Figure 2 in the paper).
> > > > > >
> > > > > > |                         | LASeR Win Rate (Llama-3-8B) | LASeR Win Rate (Mistral-7B) |
> > > > > > |----------------------------------|-----------------------------|-----------------------------|
> > > > > > | LASeR vs. Best RM                | 56.34%                     | 58.72%                     |
> > > > > > | LASeR vs. Classifier Selection   | 69.52%                     | 60.37%                     |
> > > > > > | LASeR vs. Sequential RM Selection      | 71.45%                     | 63.72%                     |
> > > > > > | LASeR vs. Random RM Selection    | 78.33%                     | 70.61%                     |
> > > > > > | LASeR vs. RM Score Ensemble      | 72.69%                     | 73.27%                     |
> > > > > > | LASeR vs. RM Agreement Ensemble     | 52.64%                     | 63.77%                     |
> > > > > >
> > > > > > * Additionally, in comparison to the RM classifier baseline that also uses training signals in the form of a trained classification head, **LASeR decisively outperforms the classifier-routed RM selection** (Table 1, Figure 2), showcasing the effectiveness of MAB-based selection. In comparison to ensemble methods, we show that our method outperforms ensemble baselines by **effectively resolving conflicts among RMs**, leading to a better training signal and performance across all tasks. This discussion is already included in lines 492-494 of the updated PDF.
> > > > > >
> > > > > > **Computational Overhead**: In our previous response, we found that added cost of covariance inverse computation is negligible relative to the overall training process. In contrast, ensemble baselines incur significantly higher overhead due to extensive loading and inference demands. For instance, ensemble methods require repeated evaluations across multiple RMs for each instance, leading to substantial computational and memory inefficiencies.
> > > > > >
> > > > > > To further clarify this point, we have revised Figure 4 in the updated PDF (lines 458-471). This figure explicitly demonstrates the accuracy-training time tradeoff for our method compared to the baselines. We observe that LASeR **not only outperforms baselines in terms of accuracy but also achieves significant training time efficiency**, making LASeR a practical and scalable solution (see table below and Figure 4).
> > > > > > | Method             | Wall-Clock Training Time (hours) | Accuracy |
> > > > > > |--------------------|-----------------------------------|--------------|
> > > > > > | Classifier Selection        | 6.55                             | 72.73        |
> > > > > > | Sequential         | 13.62                       | 72.62        |
> > > > > > | RM Agreement Ensemble    | 12.43                            | 73.85        |
> > > > > > | RM Score Ensemble     | 9.86                             | 70.94        |
> > > > > > | LASeR (ours)              | **5.49**                             | **74.75**       |
> > > > > >
> > > > > > **Broader Context and Novelty**: LASeR represents a novel and principled step toward incorporating adaptive decision-making into RM selection, a direction that has seen limited exploration in the literature. The proposed method introduces a new dimension of learning capabilities that baseline methods inherently lack. While the current results demonstrate its potential, we believe that further iterations of the approach will yield even greater efficiency and effectiveness.
> > > > > >
> > > > > > We hope this clarifies the motivations and merits of our approach and would respectfully request that you reconsider your assessment in light of these points, especially the additional results on wall-clock time. Thank you again for your valuable feedback.

---

> > > > > > > ### Author Response · Authors · 2024-11-29
> > > > > > > **Reminder to Reviewer QEwv**
> > > > > > >
> > > > > > > Dear Reviewer QEwv, we believe our response and the revisions to our paper have addressed your comments. We would appreciate your feedback on our response and updated paper and would politely ask you to revisit your score if our updated results have addressed your comments.

---

> > > > > > > > ### Author Response · Authors · 2024-12-02
> > > > > > > > **Reminder to Reviewer QEwv**
> > > > > > > >
> > > > > > > > Since today is the deadline for the authors-reviewers discussion period, we would be truly grateful if you could review our latest response. If you find that our answers address your comments, we would greatly appreciate it if you could consider raising your evaluation accordingly. Otherwise, we are happy to address any further questions you might have in the remaining time before the deadline.

---

> > > > > > > > > ### Comment · Reviewer_QEwv · 2024-12-03
> > > > > > > > >
> > > > > > > > > Thank you for your response and clarification. I still have a few concerns and decided to maintain my scores. 1) From Table 2, LASeR does not consistently outperform the other baselines. 2) Also, my main concern is whether the use of LinUCB is appropriate, due to the slight improvements shown in Tables 1 and 2 compared to the baselines. 3) I did not identify a specific challenge that LinUCB addresses in this scenario.

---

> > > > > > > > > > ### Author Response · Authors · 2024-12-04
> > > > > > > > > > **Follow-up Response to Reviewer QEwv**
> > > > > > > > > >
> > > > > > > > > > **Improvement in Table 2:** LASeR outperforms the baselines on most tasks when using LLaMA-3-8B and Mistral-7B except for summarization. For summarization, our performance is comparable to the best baseline.
> > > > > > > > > >
> > > > > > > > > > On the remaining tasks, LASeR achieves significantly larger improvements over the baselines. As mentioned previously, adaptive RM selection is particularly effective in settings where LLMs are trained on queries from multiple diverse domains, such as the instruction-following setup (lines 409–423). In a direct comparison, including additional baselines, LASeR demonstrates a substantially higher AlpacaEval win rate.
> > > > > > > > > >
> > > > > > > > > > Furthermore, we note that while Best RM baseline slightly outperforms LASeR on summarization tasks in Table 2, the same baseline underperforms on reasoning tasks in Table 2 and on instruction-following tasks in Table 3. Taken together, we believe that our claim that LASeR is the most effective method across tasks, models, and domains still holds.
> > > > > > > > > >
> > > > > > > > > > **The use of LinUCB and challenge that LASeR resolved:** The specific challenge that LinUCB addresses in this scenario is the need to balance exploration and exploitation effectively. In our setting, exploration means trying different RMs across examples to learn how well each RM performs on specific tasks (since this fine-grained information is not known apriori). For example, if we over-explore, we might waste time on RMs that don’t perform well, slowing down progress. On the other hand, if we over-exploit, we might prematurely focus on one RM that seems best initially but isn’t optimal for all queries or tasks (lines 224-227).
> > > > > > > > > > LASeR strikes the right balance because it makes RM selection an instance-level decision. It uses exploration to test various RMs on different queries. Then, it exploits this knowledge to prioritize the most suitable RM for future queries. This dynamic process ensures that LASeR adapts to the specific needs of each query while improving overall training performance.
> > > > > > > > > > We have justified this in the experiments (lines 359–366), LASeR outperforms both exploration-only and exploitation-only baselines. Exploration-only methods like random or sequential selection ignore RM performance, while exploitation-only approaches, like selecting a single RM based on aggregate scores, fail to adapt to query-specific needs. LASeR’s ability to balance these two goals explains its superior results across all tasks.
> > > > > > > > > >
> > > > > > > > > > Additionally, we observed the presence of conflicting signals among RMs (lines 472-494). LASeR addresses these conflicts more effectively than ensemble baselines because it resolves discrepancies between RMs dynamically during training. This ability to adaptively choose the most suitable RM at the instance level explains why LASeR and the "Best RM" baseline outperform multi-RM ensembles, which cannot efficiently handle these conflicts.
> > > > > > > > > >
> > > > > > > > > > We hope this addresses your concerns and will allow you to revisit your score. Thank you for engaging in discussion.

---

### Official Review · Reviewer_yAVB · 2024-11-05

**Soundness:** 2
**Presentation:** 3
**Contribution:** 2
**Rating:** 3
**Confidence:** 4

**Summary:**

The paper discusses one method to use multiple reward models during training. The method uses a linear adapter, called A_k corresponding to reward model k \in [K] which is used to obtain a certain "reward". This reward pertains to how good a reward model is for a given batch of inputs. The inputs are modelled as vectors obtained from the last token of a sentence, averaged over the sentences in the batch. The method strongly suggests using multiple reward models, but why use their method of LinUCB is not clear. The paper suggests that ensembling works, but there are many methods for ensembling. Why use their method is not clear from the given experiments.

**Strengths:**

- The results are strong. They show improvements over their chosen baselines.
- The paper is written clearly, and is highly readable. All the points are well covered.

**Weaknesses:**

- There is no baseline that selects more than 1 reward model over an epoch of training. The baselines are rather weak. What about a simple baseline that learns a classifier to choose the reward model based on the type of query. If Bandit algorithms perform better than such an approach, we can conclude that the method of using covariance works. Without any such baseline, we are left with the conclusion that one should choose the RM depending on the input c(t). Further to this, why use the authors' specific method dependent on c(t) is not clear.
- Why use MAB? why is there no discussion on the advantages gained by using the "exploration" that MABs provide?
- Adversarial training set choice. Imagine there are k RMs. Assume RM k is well coordinated with input_i where i%K = k. In other words, this RM k correlates well with human judgement for every i'th instance for i%K = k. For the remaining inputs, it performs poorly. Similarly, assume this is true for all RMs RM_i \in {RM_1,...RM_K}. Then assume a batch is of size K, where batches arrive without randomized order (i.e. sequential order). Then given a batch, no RM would correspond to the required outputs for all inputs in the batch.

- Why LinUCB? What is the "exploration" here? Why not simply train a classifier (or matrix A) to choose based on some criterion/classifier?
- Why not a linear combination of the reward functions? What if two are good? Why use arg-max over a weighted sum?

**Questions:**

- Please give more insights on each of the RMs used. For example which RM performs well for which examples? Once that is known, could you cluster the batches for training based on all inputs in the same batch using the same RM? Would that not perform better? Currently a batch is a mix of different inputs, some of which may not be suitable to the RM chosen for the batch.
- If one is using multiple RMs, would one not like to tune the RM itself instead of choosing between RMs that are designed to be good general purpose models (i.e. can you train domain specific RMs rather than generic RMs)? I know the setting is that the reward models are already given to you. But if you are using multiple RMs, would you not like to train the RMs to be good at certain domains?

### Suggestions:

- Please consider adding more details on what the RMs are, and how they were chosen. Does the method generalize to more than 4 RMs?

### Limitations:
- The experiments did not make it clear to me that this is the best way to use multiple reward models. in fact none of the baselines are allowed to choose between multiple RMs for different batches. This points to a trade-off between the memory usage of multiple RMs vs accuracy on downstream tasks.

### Additional points
Happy to raise my score given a convincing rebuttal on some of the concerns raised.

---

> ### Author Response · Authors · 2024-11-19
> **Response to Reviewer yAVB [Part 1]**
>
> We would like to thank you for your detailed questions and feedback, and are glad that you found our results “strong” and the paper to be “well-written”. We have sought to address your comments below by providing specific responses to each of your points.
>
> > [W1] There is no baseline that selects more than 1 reward model over an epoch of training. What about a simple baseline that learns a classifier...
>
> * First, we would like to clarify that three of our original baselines: Random RM Selection, Sequential RM Selection and RM Score Ensemble train the LLM using multiple RMs over different batches (lines 344-358 in the updated PDF).
> * **Training a classifier:**
>    * Ideally, given an in-distribution dataset of queries and the corresponding suitable RMs for annotating preferences, one could simply train a classifier for RM selection. However, in reality, such a fine-grained and in-distribution dataset is not generally available.
>    * Based on your suggestion, we explore an approach that uses the RewardBench data to train a classifier that maps queries to a suitable RM from a set of RMs. We take each query in the RewardBench data along with its corresponding chosen and rejected responses. We then use each RM to score these responses. The RM that assigns the correct score with the highest difference between the chosen and rejected response is selected as the RM label for that query. We train this classifier and use it to select the RM which is in turn used to train the LLM. In our experiments, we use a three-layer MLP with hidden dimensions of 2048 and 1024, and an output dimension of 4 (number of RMs), with ReLU activation in each layer.
>     * The results are reported in revised Table 1, Figure 2 of Sec 4.2, where **we find that training with LASeR improves average reasoning performance over using a trained classifier by 1.72% for Llama-3-8B and by 2.09% for Mistral-7B. On WildChat, LASeR outperforms the trained classifier by an AlpacaEval win rate of 69.52% on Llama-3-8B.** We hypothesize LASeR’s improvement over the trained classifier stems from the distribution shift in the RewardBench data and the reasoning datasets we use for the training the LLM. Unlike the classifier, LASeR learns the RM selection criterion (via MAB parameters) using in-distribution data during training, which is enabled by the fact that it is bandit-based.
>
>
> > [W2 & W4] Why use MAB? Why is there no discussion on the advantages gained by using the "exploration" that MABs provide?
>
> Thanks for your suggestion, we have incorporated it in lines 224-227 of the revised paper. We note that selecting the most suitable RM for each query $x_i$ is an *instance-level decision* dependent on the underlying query and the state of the LLM (generating the responses).  Exploration in LASeR amounts to exploring different RMs for preference labeling on different examples, ultimately enabling LASeR to dynamically identify the most suitable RM for each query. Using a bandit allows the LASeR to explore new RMs and update its information about the relevance to a query and quality of preference pairs labeled according to each RM (via the MAB reward). Without exploration, one approach would be to simply select the single RM that might perform best based on the aggregate RewardBench scores (i.e., our “Best RM” baseline, lines 314-317 in the updated PDF).  We choose LinUCB because it is a contextual bandit algorithm that takes into account context information, is easy to incorporate into our framework, and provides a good trade-off between computational efficiency and performance.
>
> However, this becomes challenging when dealing with datasets that have diverse categories such as Wildchat. As noted in lines 359-366 in the updated PDF, the best RM baseline serves as an “exploit-only” setting that only exploits the best available RM based on RewardBench (without exploring any other RMs).  On the other hand, the random and sequential selection baselines are explore-only in that they pick a new RM either randomly or via a predefined sequence irrespective of the performance of each arm (RM). Throughout Sec 4.2, we show that our method is better than these two approaches. In particular, LASeR with Llama-3-8B outperforms training with the single best RM by 1.45%. Moreover, LASeR beats LLMs trained with the best RM in the ensemble and a sequential baseline, with 56.34% and 71.45% win rates respectively on length-controlled AlpacaEval. These results highlight the exploration-exploitation tradeoff and underscoring the need to balance this with a MAB.

---

> > ### Author Response · Authors · 2024-11-19
> > **Response to Reviewer yAVB [Part 2]**
> >
> > > [W3] Adversarial training set choice. …. Then given a batch, no RM would correspond to the required outputs for all inputs in the batch.
> >
> > Your point is well-taken. As noted in footnote 3 of the updated PDF, we experimented with different batch sizes to evaluate their impact. Using a batch size of 1 (which avoids the mismatch issue you raise) yielded comparable performance to a batch size of 16 but was significantly less efficient in training the LLM due to the increased computational overhead. Based on this, we opted to use a batch size of 16 for a better trade-off between performance and efficiency.
> >
> > For datasets like Wildchat, which contain clearly defined and diverse categories, we structure the batches such that each batch consists of data belonging to a single category. This setup minimizes the risk of mismatches between the RM and the batch data, as each RM is evaluated on its most relevant data category. For reasoning datasets where such predefined categories do not exist, shuffling the data during training ensures diverse data and good training signal for LLM within each batch.
> >
> >
> > > [W5] Why not a linear combination of the reward functions? What if two are good? Why use arg-max over a weighted sum?
> >
> > We suspect you are referring to Eq. 2, where our choice of argmax is consistent with MAB setup of selecting *one* arm or RM. While multiple RMs can be used to annotate preferences (as in one of our “Ensemble RM” baselines), one of two cases is possible:
> >
> >    (i) All RMs scores agree, leaving the preference pairs unaltered and therefore having no impact on the LLM’s training or downstream performance. In this case, choosing one RM (as the MAB does) is as good as using multiple.
> >
> >    (ii) RMs disagree in scores and preference rankings: Our analysis shows that RMs often provide highly conflicting signals (Figure 5). In such cases, introducing multiple RMs in the scoring process through a weighted sum can add further noise to the preference rankings.
> >
> > Crucially, we have validated point (ii) empirically: we demonstrated in our experiments that selecting a single RM (our method) yields better performance compared to averaging scores across multiple RMs (ensemble baseline).
> >
> > > [Q1] Please give more insights on each of the RMs used. For example which RM performs well for which examples? Once that is known, could you cluster the batches …
> >
> > Thank you for the suggestion, we have added more details about each RM in Appendix A1 lines 852-862 in the revised pdf. For a breakdown of which RM is selected by LASeR for which class of queries, we refer you to the RM distribution patterns in Fig. 6 & 7 along with relevant discussion in Sec 5 lines 492-495 (updated pdf) and Appendix B (updated pdf) . Overall, we find that LASeR’s choice of RMs often aligns with the aggregate scores on RewardBench for the specific domains. Indeed, RM usage patterns could be used to collate queries across batches. However, as we mentioned in response to your comment about adversarial batching, our current data collating setup already facilitates the same RM being used for all queries in the batch.
> >
> > > [Q2] If one is using multiple RMs, would one not like to tune the RM itself instead of choosing between RMs that are designed to be good general purpose models …
> >
> > Training domain-specific RMs for every domain would be computationally expensive and is complementary to the focus of our work. Each time we switch to a new domain, we would need to retrain or fine-tune the RM. This repeated retraining adds a significant overhead, especially in settings where data spans multiple diverse domains (such as WildChat).
> >
> > Moreover, domain-specific training of RMs can lead to over-optimization, where the RM becomes too specialized to the training domain. This can reduce its generalizability, making it perform poorly on queries or data points that are even slightly outside the training domain and thus necessitating an *almost perfect classifier* that routes each query to the corresponding domain-specific RM. As we noted in our response describing our additional classifier-based baseline above, fine-grained training data for training such a classifier is scarce and in out-of-distribution settings, LASeR outperforms such a classifier-routed baseline.
> >
> > > [Q3] Please consider adding more details... Does the method generalize to more than 4 RMs?
> >
> > Please refer to details about each RM in Appendix A1 lines 852-862 (updated pdf), we choose these RMs based on their strong performance on RewardBench leaderboard at 7B scale. As for your question about the scaling LASeR to more than 4 RMs, in Appendix C lines 1080-1086 (updated pdf), we expand the candidate set of RMs with up to 4 more RMs (ranked below the original 4 RMs on RewardBench). We find that **LASeR is robust to adding these weaker RMs with the reasoning performance remaining unchanged with additional RMs (from 4 to 8)**; thereby showing that LASeR can be effectively scaled to select from a larger set of RMs.

---

> > > ### Author Response · Authors · 2024-11-22
> > > **Follow-up reminder for Reviewer yAVB**
> > >
> > > Thank you once again for your valuable feedback. We hope our response has addressed all of your questions and will allow you to revisit your score (as you mentioned in your review), otherwise we would be happy to engage further and address any further questions you might have in the remaining few days of the discussion period.

---

> > > > ### Author Response · Authors · 2024-11-24
> > > > **Reminder to reviewer yAVB**
> > > >
> > > > Since the rebuttal period is drawing to a close, with only 2 days left before the 26th, we wanted to check in again and see whether our additional experiments/positive results and responses have addressed your comments. If they have, we would appreciate if you could revisit your score (as mentioned in your review).

---

> > > > > ### Author Response · Authors · 2024-11-27
> > > > > **Reminder to reviewer yAVB**
> > > > >
> > > > > Since today is the final day for updating the PDF for our submission, we wanted to kindly check in again to see whether our rebuttal and revised paper have addressed all your concerns. If so, we would appreciate it if you could revisit your score. Otherwise, we are happy to continue discussing any remaining questions, since the rebuttal period has been extended.

---

> > > > > > ### Author Response · Authors · 2024-11-29
> > > > > > **Reminder to Reviewer yAVB**
> > > > > >
> > > > > > Dear Reviewer yAVB, we wanted to check in again to see whether our rebuttal and revised paper have addressed all your comments. If so, we would appreciate it if you could revisit your score (as mentioned in your initial review). Otherwise, we are happy to continue discussing any remaining questions, since the rebuttal period has been extended.

---

### Meta-Review · Area_Chair_Dbzf · 2024-12-21

**Metareview:**

The paper proposes using a contextual bandit algorithm to select among multiple reward models
when performing mini-batch preference finetuning of LLMs.
Empirical results shown in the paper suggest that adaptively selecting reward models for different inputs/prompts
can yield substantial benefits over baselines that commit to a single reward model, or randomly select among models.

The reviewers agreed that the paper studies a well motivated problem,
but raised persistent questions about the motivation for the solution approach (i.e. the use of bandit algorithms like LinUCB).

Consider Figure 1.
Based on the reward that is revealed to the bandit (i.e. -L_m(t)), notice that the reward can also be observed for all the other arms not selected by the bandit algorithm.
For all of the responses generated by the LLM, each of the RMs can be used to create preference pairs and thereby compute -L_m(t) for each one of them.
This means that this is an online learning problem (learning from expert advice), with very effective algorithms like HEDGE, MULTIPLICATIVE WEIGHTS, etc.

**Additional Comments On Reviewer Discussion:**

During the rebuttal, the authors added baselines to their experiments with offline classifiers trained on a different dataset (and found that the distribution shift made such classifiers not work as well as the bandit algorithm learned in-distribution).
Adding the learning-from-expert-advice baselines will substantially strengthen the claimed benefits of exploration from the contextual bandit algorithm.

---

### Decision · Program_Chairs · 2025-01-22

Reject